# Contrasting action and posture coding with hierarchical deep neural network models of proprioception

Kai J Sandbrink[1†§], Pranav Mamidanna[2†#], Claudio Michaelis[2], Matthias Bethge[2], Mackenzie Weygandt Mathis[1,3*‡], Alexander Mathis[1,3*‡]

[1]The Rowland Institute at Harvard, Harvard University, Cambridge, United States; [2]Tübingen AI Center, Eberhard Karls Universität Tübingen & Institute for Theoretical Physics, Tübingen, Germany; [3]Brain Mind Institute, School of Life Sciences, École Polytechnique Fédérale de Lausanne, Genève, Switzerland

*For correspondence:
mackenzie@post.harvard.edu
(MWM);
alexander.mathis@epfl.ch (AM)

†These authors contributed
equally to this work
‡These authors also contributed
equally to this work

Present address: §Department
of Experimental Psychology,
University of Oxford, Oxford,
United Kingdom; #Department of
Health Science and Technology,
Aalborg University, Aalborg,
Denmark

Competing interest: See page
25

Reviewing Editor: Demba Ba,
Harvard University, United States

**Abstract** Biological motor control is versatile, efficient, and depends on proprioceptive feedback. Muscles are flexible and undergo continuous changes, requiring distributed adaptive control mechanisms that continuously account for the body's state. The canonical role of proprioception is representing the body state. We hypothesize that the proprioceptive system could also be critical for high-level tasks such as action recognition. To test this theory, we pursued a task-driven modeling approach, which allowed us to isolate the study of proprioception. We generated a large synthetic dataset of human arm trajectories tracing characters of the Latin alphabet in 3D space, together with muscle activities obtained from a musculoskeletal model and model-based muscle spindle activity. Next, we compared two classes of tasks: trajectory decoding and action recognition, which allowed us to train hierarchical models to decode either the position and velocity of the end-effector of one's posture or the character (action) identity from the spindle firing patterns. We found that artificial neural networks could robustly solve both tasks, and the networks' units show tuning properties similar to neurons in the primate somatosensory cortex and the brainstem. Remarkably, we found uniformly distributed directional selective units only with the action-recognition-trained models and not the trajectory-decoding-trained models. This suggests that proprioceptive encoding is additionally associated with higher-level functions such as action recognition and therefore provides new, experimentally testable hypotheses of how proprioception aids in adaptive motor control.

## Editor's evaluation

This article proposes a combination of biomechanical modeling and in-silico experiments, on a newly-curated passive-movement dataset, to elucidate the nature of computations in the proprioceptive pathway. The authors find that, in addition to its canonical role in representing the body state, the proprioceptive pathway may have evolved to recognize actions. Overall, the authors' findings lead to new hypotheses about proprioception that future in-vivo experiments could test.

## Introduction

Proprioception is a critical component of our ability to perform complex movements, localize our body's posture in space, and adapt to environmental changes (*Miall et al., 2018*; *Proske and Gandevia, 2012*; *Delhaye et al., 2018*). Our movements are generated by a large number of muscles and are sensed via a diverse set of receptors, most importantly muscle spindles, which carry highly multiplexed information (*Clark et al., 1985*; *Proske and Gandevia, 2012*). For instance, arm movements

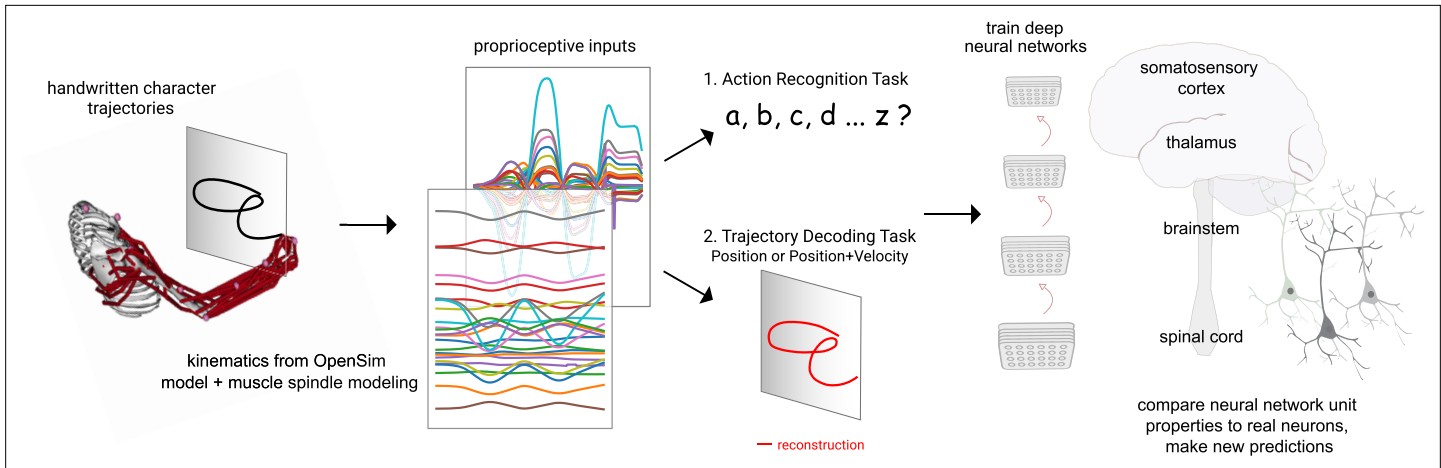

**Figure 1.** Contrasting spindle-based tasks to study proprioception: proprioceptive inputs that correspond to the tracing of individual letters were simulated using a musculoskeletal model of a human arm. This scalable, large-scale dataset was used to train deep neural network models of the proprioceptive pathway either to classify the character (action recognition task [ART]) or to decode the posture of the arm (trajectory decoding tasks [TDTs]) based on the input muscle spindle firing rates. Here, we test two variants of the latter, to decode either position-only (canonical-proprioception) or position+velocity (as a control) of the end-effector. We then analyze these models and compare their tuning properties to the proprioceptive system of primates.

are sensed via distributed and individually ambiguous activity patterns of muscle spindles, which depend on relative joint configurations rather than the absolute hand position (*Matthews, 1963*; *Matthews, 1981*). Interpreting this high-dimensional input (around 50 muscles for a human arm) of distributed information at the relevant behavioral level poses a challenging decoding problem for the central nervous system (*Bernstein, 1967*; *Matthews, 1981*). Proprioceptive information from the receptors undergoes several processing steps before reaching somatosensory cortex (*Bosco et al., 1996*; *Delhaye et al., 2018*; *Tuthill and Azim, 2018*) – from the spindles that synapse in Clarke's nucleus, to the brainstem, thalamus (*Francis et al., 2008*; *Delhaye et al., 2018*), and finally to somatosensory cortex (S1). In cortex, a number of tuning properties have been observed, such as responsiveness to varied combinations of joints and muscle lengths (*Goodman et al., 2019*; *Chowdhury et al., 2020*), sensitivity to different loads and angles (*Fromm and Evarts, 1982*), and broad and uni-modal tuning for movement direction during arm movements (*Prud'homme and Kalaska, 1994*). The proprioceptive information in S1 is then hypothesized to serve as the basis of a wide variety of tasks via its connections to motor cortex and higher somatosensory processing regions (*Miall et al., 2018*; *Proske and Gandevia, 2012*; *Delhaye et al., 2018*; *Mathis et al., 2017*; *Kumar et al., 2019*).

One key role of proprioception is to sense the state of the body, that is, posture. This information subserves many other functions, from balance to motor learning. Thus, to gain insights into the computations of the proprioceptive system, we quantitatively compare two different goals in a task-driven fashion: a trajectory-decoding task and an action recognition task (ART) (*Figure 1*). The trajectory-decoding task represents the canonical view of proprioception (*Proske and Gandevia, 2012*; *Delhaye et al., 2018*). Alternatively, the role of the proprioceptive system might include inference of more abstract actions (i.e., complex sequences of postures). Our hypothesis is motivated by the observation that action segmentation would be an efficient way to represent complex behavior, and it could directly drive the action map in motor cortex (*Graziano, 2016*). These two tasks also represent two different extremes for learning invariances: the trajectory-decoding task enforces invariance to 'what' is done, while the ART encourages invariance to 'where' something is done. Along this continuum, we also consider a variant of trajectory decoding which also predicts velocity. The ART is also motivated by the following observation: although the animal's motor system is aware of its own actions (at least during volitional control), it may still be helpful to infer executed actions in order to direct corrective motor actions in the event of disturbances (*Todorov and Jordan, 2002*; *Mathis et al., 2017*) or to serve as targets for action reinforcement (*Markowitz et al., 2023*).

Large-scale datasets like ImageNet (*Russakovsky et al., 2015*), that present a challenging visual object-recognition task, have allowed the training of deep neural networks whose representations

closely resemble the tuning properties of single neurons in the ventral pathway of primates (*Khaligh-Razavi and Kriegeskorte, 2014*; *Yamins et al., 2014*; *Cichy et al., 2016*; *Yamins and DiCarlo, 2016*; *Schrimpf et al., 2018*; *Cadena et al., 2019*; *Storrs et al., 2021*). This goal-driven modeling approach (*Yamins and DiCarlo, 2016*; *Richards et al., 2019*; *Saxe et al., 2020*) has since successfully been applied to other sensory modalities such as touch (*Zhuang et al., 2017*; *Sundaram et al., 2019*), thermosensation (*Haesemeyer et al., 2019*), and audition (*Kell et al., 2018*). However, unlike for vision and audition, where large annotated datasets of raw images or sounds are readily available, data for relevant proprioceptive stimuli (as well as task goals) are not.

To create a large-scale passive movement dataset, we started with human motion data for drawing different Latin characters (*Williams et al., 2006*). Next, we used a musculoskeletal model of the human upper limb (*Saul et al., 2015*) to generate muscle length configurations corresponding to drawing the pen-tip trajectories in multiple horizontal and vertical planes. We converted these into proprioceptive inputs using models of spindle Ia and II (*Dimitriou and Edin, 2008a*; *Dimitriou and Edin, 2008b*). We then used the tasks to train families of neural networks to either decode the full trajectory of the handwritten characters or classify the characters from the generated spindle firing rates. Through an extensive hyper-parameter search, we found neural networks for various architectures that solve the tasks. We then analyzed those models and found that models trained on action recognition, but not trajectory decoding, more closely resemble what is known about tuning properties in the proprioceptive pathway. Collectively, we present a framework for studying the proprioceptive pathway using goal-driven modeling by synthesizing datasets of muscle (spindle) activities in order to test theories of coding.

As in previous task-driven work for other sensory systems (*Yamins and DiCarlo, 2016*; *Kell et al., 2018*), we do not model the system in a closed-loop nature with a motor control model. Of course, tuning properties of proprioception are likely optimized jointly with the motor system (and possibly other systems). Studying proprioception with a basic open-loop model is important to (1) isolate proprioception, and (2) set the stage for comparing to more complex models, such as joint models of proprioception and motor control.

## Results

### Muscle spindle-based biomechanical tasks

To model the proprioceptive system, we designed two classes of real-world proprioceptive tasks. The objectives were to either classify or reconstruct Latin alphabet characters (character recognition or trajectory decoding) based on the proprioceptive inputs that arise when the arm is passively moved (*Figure 1*). Thus, we computationally isolate proprioception from active movement, a challenge in experimental work. We used a dataset of pen-tip trajectories for the 20 characters that can be handwritten in a single stroke (thus excluding *f, i, j, k, t*, and *x*, which are multi-stroke) (*Williams, 2008*; *Williams et al., 2006*). Then, we generated 1 million end-effector (hand) trajectories by scaling, rotating, shearing, translating, and varying the speed, of each original trajectory (*Figure 2A–C*; *Table 1*).

To translate end-effector trajectories into three-dimensional (3D) arm movements, we computed the joint-angle trajectories through inverse kinematics using a constrained optimization approach (*Figure 2D–E* and Methods). We iteratively constrained the solution space by choosing joint angles in the vicinity of the previous configuration in order to eliminate redundancy. To cover a large 3D workspace, we placed the characters in multiple horizontal (26) and vertical (18) planes and calculated corresponding joint-angle trajectories (starting points are illustrated in *Figure 2D*). A human upper-limb model in OpenSim (*Saul et al., 2015*) was then used to compute equilibrium muscle lengths for 25 muscles in the upper arm that lead to the corresponding joint-angle trajectory (*Figure 2F*, *Figure 2—video 1*). We did not include hand muscles for simplicity, therefore the location of the end-effector is taken to be the hand location.

Based on these simulations, we generated proprioceptive inputs composed of muscle length and muscle velocity, which approximate receptor inputs during passive movement (see Methods). From this set, we selected a subset of 200,000 examples with smooth, non-jerky joint angle and muscle length changes, while ensuring that the set is balanced in terms of the number of examples per class (see Methods). Since not all characters take the same amount of time to write, we padded the

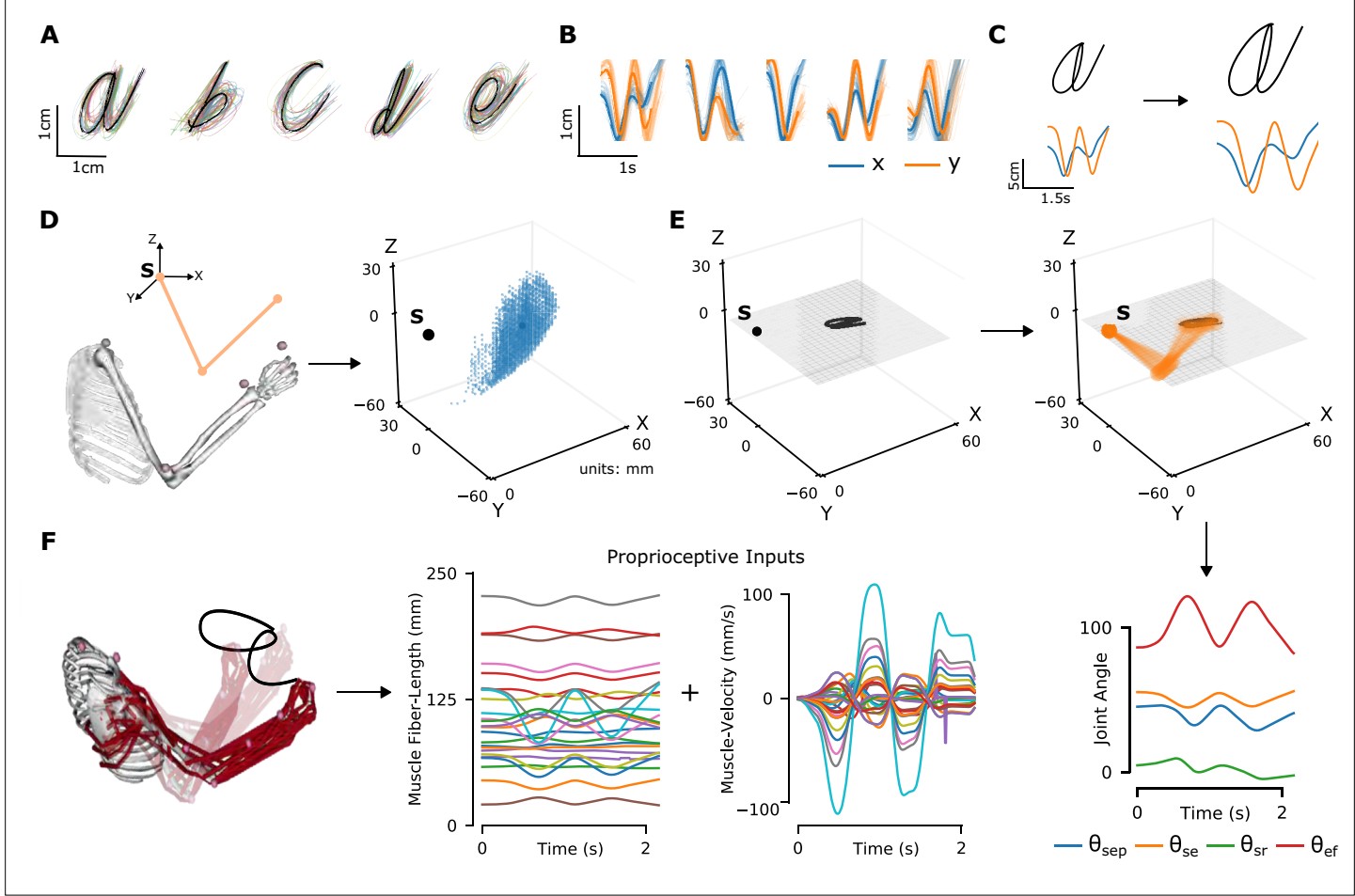

**Figure 2.** Synthetic proprioceptive characters dataset generation. (**A**) Multiple examples of pen-tip trajectories for five of the 20 letters are shown. (**B**) The same trajectories as in A, plotted as time courses of Cartesian coordinates. (**C**) Creating (hand) end-effector trajectories from pen-tip trajectories. (*Left*) An example trajectory of character a resized to fit in a 10 × 10 cm² grid, linearly interpolated from the true trajectory while maintaining the true velocity profile. (*Right*) This trajectory is further transformed by scaling, rotating, and varying its speed. (**D**) Candidate starting points to write the character in space. (*Left*) A 2-link, 4 degrees of freedom (DoFs) model human arm is used to randomly select several candidate starting points in the workspace of the arm (*right*), such that written characters are all strictly reachable by the arm. (**E**) (*Left to right and down*) Given a sample trajectory in (C) and a starting point in the arm's workspace, the trajectory is then drawn on either a vertical or horizontal plane that passes through the starting point. We then apply inverse kinematics to solve for the joint angles required to produce the traced trajectory. (**F**) (*Left to right*) The joint angles obtained in (E) are used to drive a musculoskeletal model of the human arm in OpenSim, to obtain equilibrium muscle fiber-length trajectories of 25 relevant upper arm muscles. These muscle fiber lengths and their instantaneous velocities together form the proprioceptive inputs.

The online version of this article includes the following video for figure 2:

**Figure 2—video 1.** Supplementary video.

https://elifesciences.org/articles/81499/figures#fig2video1

movements with static postures corresponding to the starting and ending postures of the movement and randomized the initiation of the movement in order to maintain ambiguity about when the writing begins. At the end of this process, each sample consists of simulated proprioceptive inputs from each of the 25 muscles over a period of 4.8 s, simulated at 66.7 Hz. The dataset was split into a training, validation, and test set with a 72-8-20 ratio.

## Recognizing characters from muscle activity is challenging

We reasoned that several factors complicate the recognition of a specific character. First, the end-effector position is only present as a distributed pattern of muscle activity. Second, the same character will give rise to widely different proprioceptive inputs depending on different arm configurations.

To test these hypotheses, we first visualized the data at the level of proprioceptive inputs by using t-distributed stochastic neighbor embedding (t-SNE, *Maaten and Hinton, 2008*). This illustrated that character identity was indeed entangled (*Figure 3A*). Then, we trained pairwise support vector machine (SVM) classifiers as baseline models for character recognition. Here, the influence of the specific geometry of each character is notable. On average, the pairwise accuracy is 86.6 ± 12.5 (mean ± SD, $N = 190$ pairs, *Figure 3B*). As expected, similar-looking characters were harder to distinguish at the level of the proprioceptive input, that is *e* and *y* were easily distinguishable but *m* and *w* were not (*Figure 3B*).

To quantify the separability between all characters, we used a one-against-one strategy with trained pairwise classifiers (*Hsu and Lin, 2002*). The performance of this multi-class decoder was poor regardless of whether the input was end-effector coordinates, joint angles, normalized muscle lengths, or proprioceptive inputs (*Figure 3C*). Taken together, these analyses highlight that it is difficult to extract the character class from those representations as illustrated by t-SNE embedding (*Figure 3A*) and quantified by SVMs (*Figure 3B, C*). In contrast, as expected, accurately decoding the end-effector position (by linear regression) from the proprioceptive input is much simpler, with an average decoding error of 1.72 cm, in a 3D workspace approximately 90 × 90 × 120 cm$^3$ (*Figure 3C*).

## Neural networks models of proprioception

We explore the ability of three types of artificial neural network models (ANNs) to solve the proprioceptive character recognition and decoding tasks. ANNs are powerful models for both their performance and for their ability to elucidate neural representations and computations (*Yamins and DiCarlo, 2016*; *Hausmann et al., 2021*). An ANN consists of layers of simplified units (neurons) whose connectivity patterns mimic the hierarchical, integrative properties of biological neurons and anatomical pathways (*Rumelhart et al., 1986*; *Yamins and DiCarlo, 2016*; *Richards et al., 2019*). As candidate models we parameterized a spatial-temporal convolutional neural network, a spatiotemporal convolutional network (both TCNs; *Lecun et al., 1998*), and a recurrent neural network (a long short-term memory [LSTM] network; *Hochreiter and Schmidhuber, 1997*), which impose different inductive priors on the computations. We refer to these three types as spatial-temporal, spatiotemporal, and LSTM networks (*Figure 3D*).

Importantly, the different models differ in the way they integrate spatial and temporal information along the hierarchy. These two types of information can be processed either sequentially, as is the case for the spatial-temporal network type that contains layers with one-dimensional (1D) filters that first integrate information across the different muscles, followed by an equal number of layers that integrate only in the temporal dimension, or simultaneously, using two-dimensional (2D) kernels, as they are in the spatiotemporal network. In the LSTM networks, spatial information was integrated similarly to the spatial-temporal networks, before entering the LSTM layer.

Candidate models for each class can be created by varying hyper-parameters such as the number of layers, number and size of spatial and temporal filters, type of regularization, and response normalization (see *Table 2*, Methods). As a first step to restrict the number of models, we performed a hyper-parameter architecture search by selecting models according to their performance on the proprioceptive tasks. We should emphasize that our ANNs integrate along both proprioceptive inputs and time, unlike standard feed-forward CNN models of the visual pathway that just operate on images (*Yamins and DiCarlo, 2016*). TCNs have been shown to be excellent for time-series modeling (*Bai et al., 2018*), and therefore naturally describe neurons along a sensory pathway that integrates spatiotemporal inputs.

## Architecture search and representational changes

To find models that could solve the proprioceptive tasks, we performed an architecture search and trained 150 models (50 models per type). Notably, we trained the same model (as specified by architectural parameters) on both classes of tasks by modifying the output and the loss function used to train the model. After training, all models were evaluated on an unseen test set (*Figure 3—figure supplement 1A*).

Models of all three types achieved excellent performance on the ART (*Figure 3E*; multi-class accuracy of 98.86% ± 0.04, mean ± SEM for the best spatial-temporal model, 97.93% ± 0.03 for the best spatiotemporal model, and 99.56% ± 0.04 for the best LSTM model, $N = 5$ randomly initialized

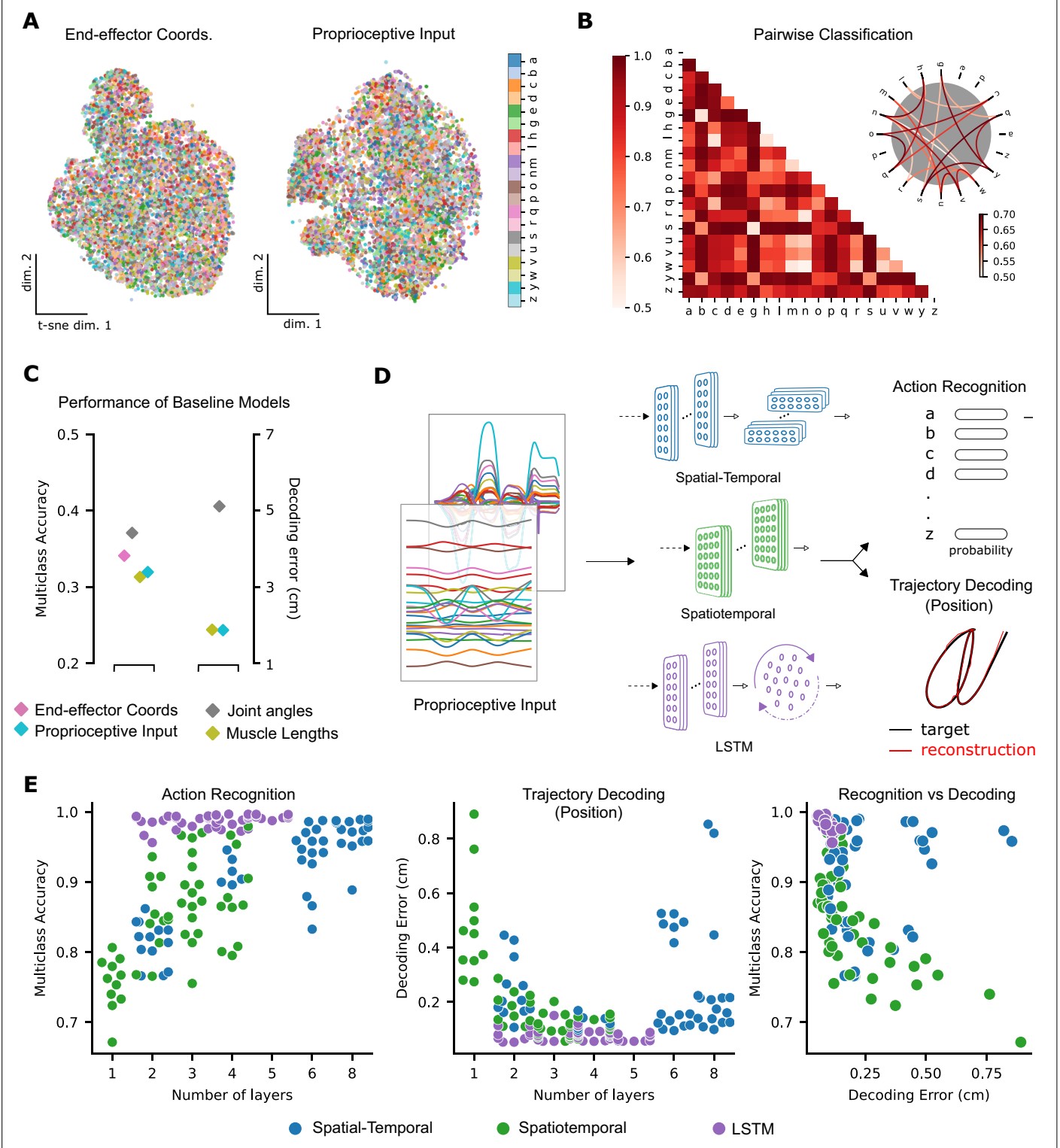

**Figure 3.** Quantifying action recognition and trajectory decoding task performance. (**A**) t-Distributed stochastic neighbor embedding (t-SNE) of the end-effector coordinates (left) and proprioceptive inputs (right). (**B**) Classification performance for all pairs of characters with binary support vector machines (SVMs) trained on proprioceptive inputs. Chance level accuracy is 50%. The pairwise accuracy is 86.6 ± 12.5% (mean ± SD, $N = 190$ pairs). A subset of the data is also illustrated as a circular graph, whose edge color denotes the classification accuracy. For clarity, only pairs with performance less than 70% are shown, which corresponds to the bottom 12% of all pairs. (**C**) Performance of baseline models: Multi-class SVM performance computed using a one-vs-one strategy for different types of input/kinematic representations on the action recognition task (left). Performance of ordinary least-

*Figure 3 continued on next page*

*Figure 3 continued*

squares linear regression on the trajectory decoding (position) task (right). Note that end-effector coordinates, for which this analysis is trivial, are excluded. (**D**) Neural networks are trained on two main tasks: action recognition and trajectory decoding (of position) based on proprioceptive inputs. We tested three families of neural network architectures. Each model is comprised of one or more processing layers, as shown. Processing of spatial and temporal information takes place through a series of one-dimensional (1D) or two-dimensional (2D) convolutional layers or a recurrent layer. (**E**) Performance of neural network models on the tasks: the test performance of the 50 networks of each type is plotted against the number of layers of processing in the networks for the action recognition (left) and trajectory decoding (center) tasks separately and against each other (right). Note we jittered the number of layers (*x*-values) for visibility, but per model it is discrete.

The online version of this article includes the following figure supplement(s) for figure 3:

**Figure supplement 1.** Network performance.

models). The parameters of the best-performing architectures are displayed in *Table 2*. The same models could also accurately solve the trajectory decoding task (TDT; position decoding) (*Figure 3E*; with decoding errors of only 0.22 cm ± 0.005, mean ± SEM for the best spatial-temporal model, 0.13 cm ± 0.005 for the best spatiotemporal model, and 0.05 cm ± 0.01 for the best LSTM model, $N = 5$ randomly initialized models). This decoding error is substantially lower than the linear readout (*Figure 3C*). Of the hyper-parameters considered, the depth of the networks influenced performance the most (*Figure 3E*, *Figure 3—figure supplement 1B*). Further, the performance on the two tasks was related: models performing well on one task tend to perform well on the other (*Figure 3E*).

Having found models that robustly solve the tasks we sought to analyze their properties. We created five pre-training (untrained) and post-training (trained) pairs of models for the best-performing model architecture for further analysis. We will refer to those as instantiations. As expected, the untrained models performed at chance level (5%) on the ART.

How did the population activity change across the layers after learning the tasks? Here, we focus on the best spatial-temporal model and then show that our analysis extends to the other model types. We compared the representations across different layers for each trained model to its untrained counterpart by linear centered kernel alignment (CKA, see Methods). This analysis revealed that for all instantiations, the representations remained similar between the trained and untrained models for the first few layers and then deviated in the middle to final layers of the network (*Figure 4A*). Furthermore, trained models not only differed from the untrained ones but also across tasks, and the divergence appeared earlier (*Figure 4A*). Therefore, we found that both training and the task substantially changed the representations. Next, we aimed to understand how the tasks are solved, that is, how the different stimuli are transformed across the hierarchy.

To illustrate the geometry of the ANN representations and how the different characters are disentangled across the hierarchy, we used t-SNE to visualize the structure of the hidden layer representations. For the ART, the different characters separate in the final layers of the processing hierarchy (spatial-temporal model: *Figure 4B*; for the other model classes, see *Figure 4—figure supplement 1A*). To quantify this, we computed representational dissimilarity matrices (RDMs; see Methods). We found that different instances of the same characters were not represented similarly at the level of proprioceptive inputs, but rather at the level of the last convolutional layer for the trained models (*Figure 4C*; for other model classes, see *Figure 4—figure supplement 1B*). To quantify how the characters are represented across the hierarchy, we computed the similarity to an Oracle's RDM, where an Oracle (or ideal observer) would have a block structure, with dissimilarity 0 for all stimuli of the same class and 1 (100th percentile) otherwise (*Figure 4—figure supplement 1B*). We found for all model instantiations that similarity only increased toward the last layers (*Figure 4D*). This finding corroborates the visual impression gained via t-SNE that different characters are disentangled near the end of the processing hierarchy (*Figure 4B*, *Figure 4—figure supplement 1A, C*).

How is the TDT solved across the hierarchy? In contrast to the ART-trained models, as expected, representations of characters remained entangled throughout (*Figure 4B*, *Figure 4—figure supplement 1A*). We found that the end-effector position can be decoded across the hierarchy (*Figure 4E*). This result is expected, as even from the proprioceptive input a linear readout achieves good performance (*Figure 3C*). We quantified CKA scores across the different architecture classes and found that with increasing depth the representations diverge between the two tasks (*Figure 4F*, *Figure 4—figure supplement 1C*). Collectively, this suggests that characters are not immediately separable in ART models, but the end-effector can be decoded well in TDT models throughout the architecture.

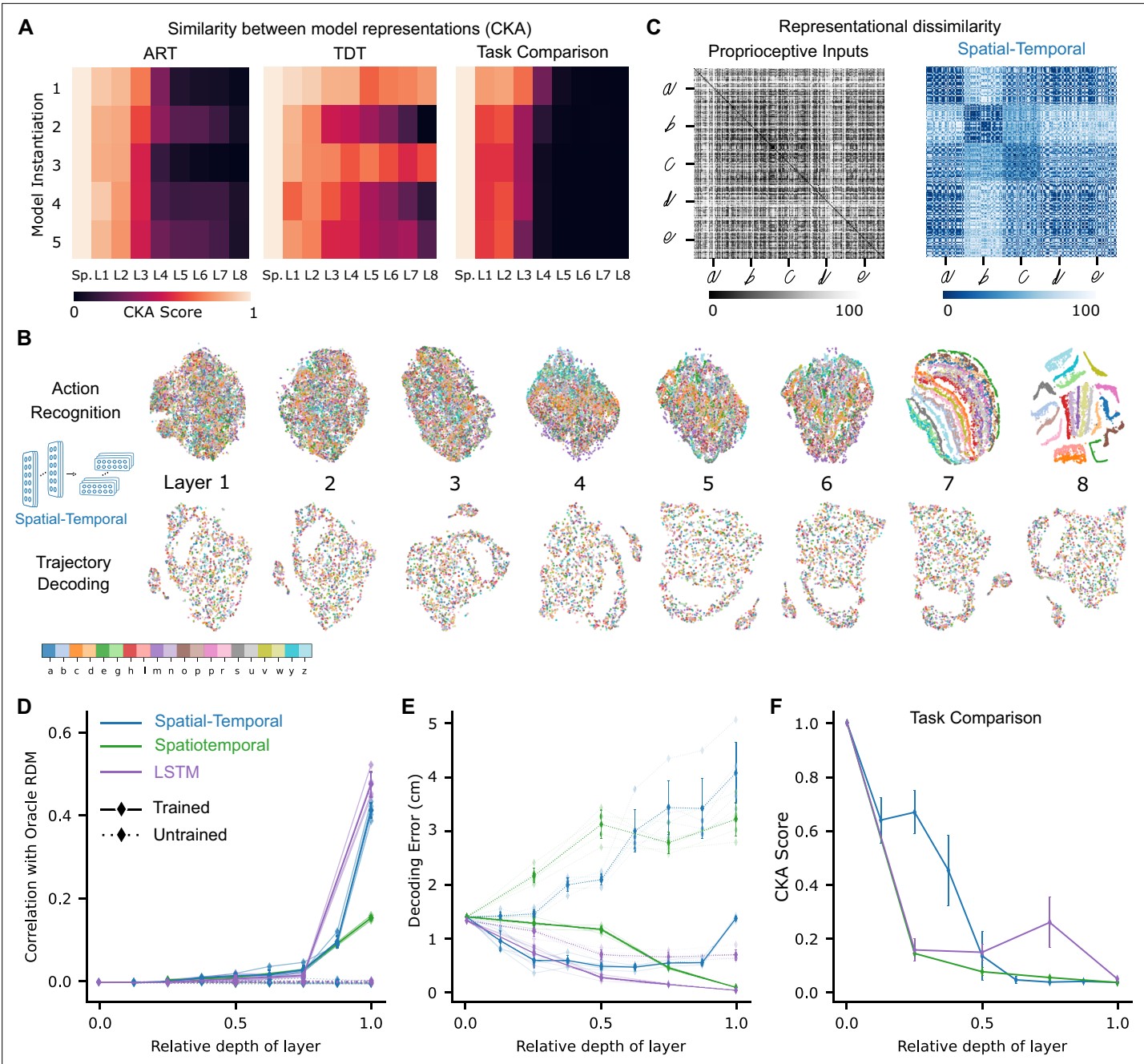

**Figure 4.** Low-dimensional embedding of network layers reveals structure. (**A**) Similarity in representations (centered kernel alignment [CKA]) between the trained and the untrained models for each of the five instantiations of the best-performing spatial-temporal models (left and center). CKA between models trained on recognition vs. decoding (right). (**B**) t-Distributed stochastic neighbor embedding (t-SNE) for each layer of one instantiation of the best-performing spatial-temporal model trained on both tasks. Each data point is a random stimulus sample (*N*=2000, 50 per stimulus). (**C**) Representational dissimilarity matrices (RDMs). Character level representation is visualized using percentile RDMs for proprioceptive inputs (left) and final layer features (right) of one instantiation of the best-performing spatio-temporal model trained on the recognition task. (**D**) Similarity in stimulus representations between RDMs of an Oracle (ideal observer) and each layer for the five instantiations of the action recognition task (ART)-trained models and their untrained counterparts. (**E**) Decoding error (in cm) along the hierarchy for each model type on the trajectory decoding task. (**F**) Centered kernel alignment (CKA) between models trained on recognition vs. decoding for the five instantiations of all network types (right).

The online version of this article includes the following figure supplement(s) for figure 4:

**Figure supplement 1.** Extended analysis of network models.

## Single unit encoding properties and decodability

To gain insight into why ART- and TDT-trained models differ in their representations, we examined single unit tuning properties. In primates, these have been described in detail (*Prud'homme and Kalaska, 1994*; *Delhaye et al., 2018*), and thus present an ideal point of comparison. Specifically, we analyzed the units for end-effector position, speed, direction, velocity, and acceleration tuning. We performed these analyses by relating variables (such as movement direction) to the activity of single units during the continuous movement (see Methods). Units with a test $R^2 > 0.2$ were considered tuned to that feature (this is a conservative value in comparison to experimental studies, e.g., 0.07 for *Prud'homme and Kalaska, 1994*).

Given the precedence in the literature, we focused on direction tuning in *all horizontal* planes. We fit directional tuning curves to the units with respect to the instantaneous movement direction. As illustrated in examples, the ART spatial-temporal model (as well as proprioceptive inputs), directional tuning can be observed for the typical units shown (*Figure 5A and B*). Spindle afferents are known to be tuned to motion, that is velocity and direction (*Ribot-Ciscar et al., 2003*). We verified the tuning of the spindles and found that the spindle component tuned for muscle length is primarily tuned for position (median $R^2 = 0.36$, $N = 25$) rather than kinematics (median direction $R^2 = 0.001$, median velocity $R^2 = 0.0026$, $N = 25$), whereas the spindle component tuned for changes in muscle length were primarily tuned for kinematics (median direction $R^2 = 0.58$, velocity $R^2 = 0.83$, $N = 25$), and poorly tuned for position (median $R^2 = 0.0024$, $N = 25$). For the ART model, direction selectivity was prominent in middle layers 1–6 before decreasing by layer 8, and a fraction of units exhibited tuning to other kinematic variables with $R^2 > 0.2$ (*Figure 5C*, *Figure 5—figure supplement 1A, B*).

In contrast, for the TDT model, no directional tuning was observed, but positional tuning was (*Figure 5D*). These observations are further corroborated when comparing the distributions of tuning properties (*Figure 5E*) and 90% quantiles for all the instantiations (*Figure 5F*). The difference in median tuning score between the two differently trained groups of models across the five model instantiations becomes significant starting in the first layer for both direction and position [Direction: (layer 1 $t(4) = 10.44$, p=0.0005; layer 2 $t(4) = 17.15$, p=6.78e-05; layer 3 $t(4) = 41.46$, p=2.02e-06; layer 4 $t(4) = 37.63$, p=2.98e-06; layer 5 $t(4) = 25.05$, p=1.51e-06; layer 6 $t(4) = 14.32$, p=0.0001; layer 7 $t(4) = 3.47$, p=0.026; layer 8 $t(4) = 7.61$, p=0.0016); Position: (layer 1 $t(4) = -10.00$, p=0.0006; layer 2 $t(4) = -24.62$, p=1.62e-05; layer 3 $t(4) = -19.15$, p=4.38e-05; layer 4 $t(4) = -13.08$, p=0.0002; layer 5 $t(4) = -21.57$, p=2.73e-05; layer 6 $t(4) = -57.55$, p=5.46e-07; layer 7 $t(4) = -20.80$, p=3.16e-05; layer 8 $t(4) = -16.08$, p=8.76–05)].

Given that the ART models are trained to recognize characters, we asked if single units are well tuned for specific characters. To test this, we trained an SVM to classify characters from the single unit activations. Even in the final layer (before the readout) of the spatial-temporal model, the median classification performance over the five model instantiations as measured by the normalized area under the ROC curve-based selectivity index for single units was 0.210 ± 0.006 (mean ± SEM, $N = 5$ instantiations), and was never higher than 0.40 for any individual unit across all model instantiations (see Methods). Thus, even in the final layer, there are effectively no single-character-specific units. Of course, combining the different units of the final fully connected layer gives a high-fidelity readout of the character and allows the model to achieve high classification accuracy. Thus, character identity is represented in a distributed way. In contrast, and as expected, character identity is poorly encoded in single cells for the TDT model (*Figure 5D and F*). These main results also hold for the other architecture classes (*Figure 5—figure supplement 2*, *Figure 5—figure supplement 3*).

In spatiotemporal models, in which both spatial and temporal processing occurs between all layers, we observe a monotonic decrease in the directional tuning across the four layers for the ART task and a quick decay for the TDT task (*Figure 5—figure supplement 4A, B*). Speed and acceleration tuning are present in the ART, but not in the TDT models (*Figure 5—figure supplement 1B, C*). Conversely, we find that positional coding is stable for TDT models and not the ART models. The same results hold true for LSTM models (*Figure 5—figure supplement 1E, F* and *Figure 5—figure supplement 4C, D*). The differences in directional and Cartesian positional tuning were statistically significant for all layers according to a paired t-test with 4 degrees of freedom (DoFs) for both model types. Thus, for all architecture classes, we find that strong direction selective tuning is present in early layers of models trained with the ART task, but not the TDT task.

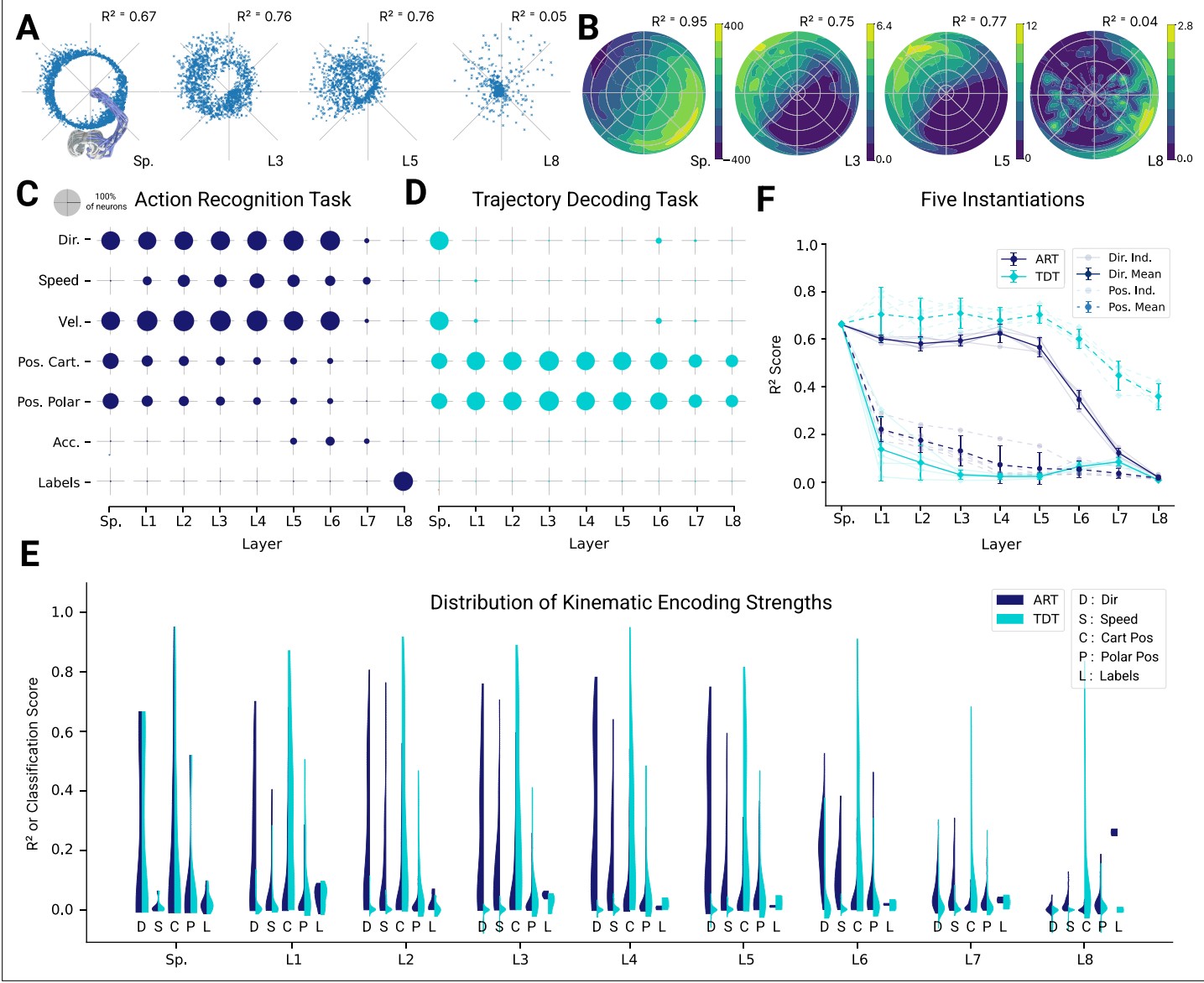

**Figure 5.** Analysis of single unit tuning properties for spatial-temporal models. (**A**) Polar scatter plots showing the activation of units (radius $r$) as a function of end-effector direction, as represented by the angle $\theta$ for directionally tuned units in different layers of the top-performing spatial-temporal model trained on the action recognition task, where direction corresponds to that of the end-effector while tracing characters in the model workspace. The activation strengths of one (velocity-dependent) muscle spindle one unit each in layers 3, 5, and 8 are shown. (**B**) Similar to (A), except that now radius describes velocity and color represents activation strength. The contours are determined following linear interpolation, with gaps filled in by neighbor interpolation and smoothed using a Gaussian filter. Examples of one muscle spindle, one unit each in layers 3, 5, and 8, are shown. (**C**) For each layer of one trained instantiation, the units are classified into types based on their tuning. A unit was classified as belonging to a particular type if its tuning had a test $R^2 > 0.2$. Tested features were direction tuning, speed tuning, velocity tuning, Cartesian and polar position tuning, acceleration tuning, and label tuning (18/5446 scores excluded for action recognition task [ART]-trained, 430/5446 for trajectory decoding task [TDT]-trained; see Methods). (**D**) The same plot but for the spatial-temporal model of the same architecture but trained on the trajectory decoding task. (**E**) For an example instantiation, the distribution of test $R^2$ scores for both the ART- and TDT-trained models are shown as vertical histograms (split-violins), for five kinds of kinematic tuning for each layer: direction tuning, speed tuning, Cartesian position tuning, polar position tuning, and label specificity indicated by different shades and arranged left-right for each layer including spindles. Tuning scores were excluded if they were equal to 1, indicating a constant neuron, or less than −0.1, indicating an improper fit (12/3890 scores excluded for ART, 285/3890 for TDT; see Methods). (**F**) The means of 90% quantiles over all five model instantiations of models trained on ART and TDT are shown for direction tuning (dark) and position tuning (light). 95% confidence intervals are shown over instantiations ($N = 5$).

The online version of this article includes the following figure supplement(s) for figure 5:

**Figure supplement 1.** Extended kinematic tuning of single neurons.

*Figure 5 continued on next page*

Our results suggest that the primate proprioceptive pathway is consistent with the action recognition hypothesis, but to corroborate this, we also assessed decoding performance, which measures representational information. For all architecture types, movement direction and speed can be better decoded from ART than from TDT-trained models (*Figure 6A, C, E*). In contrast, for all architectures, position can be better decoded for TDT- than for ART-trained models (*Figure 6B, D, F*). These results are consistent with the single-cell encoding results and again lend support for the proprioceptive system's involvement in action representation.

So far, we have directly compared TDT and ART models. This does not address task training as such. Namely, we found directional selective units in ART models and positional-selective units in TDT models, but how do those models compare to randomly initialized (untrained) models? Remarkably, directional selectivity is similar for ART-trained and untrained models (*Figure 5—figure supplement 2*). In contrast to untrained models, ART-trained models have fewer positionally tuned units. The situation is reversed for TDT-trained models – those models gain positionally tuned units and lose directionally selective units during task training (*Figure 5—figure supplement 3*). Consistent with those encoding results, position can be decoded less well from ART-trained models than from untrained models, and direction and speed similarly well (*Figure 6—figure supplement 1*). Conversely, direction and speed can be worse and position better decoded from TDT-trained than untrained models (*Figure 6—figure supplement 2*).

We found that while TDT models unlearn directional selective units (*Figure 5—figure supplement 3*), ART models retain them (*Figure 5—figure supplement 2*). As an additional control, we wanted to test a task that does not only predict the position of the end-effector but also the velocity. We therefore trained the best five instantiations of all three model types on the position and velocity trajectory decoding task (TDT-PV) task (see Methods). We found that they could accurately predict both the location and the velocity (root mean squared error: 0.11, mean ± SEM for the best spatial-temporal model, 0.09 for the best spatiotemporal model, and 0.04 for the best LSTM model, $N = 5$ best models) and produced similar position decoding errors as the positional TDT-trained models (decoding errors for the TDT-PV-trained models: 0.26 cm ± 0.01, for the best spatial-temporal model, 0.20 cm ± 0.01 for the best spatiotemporal model, and 0.09 cm ±0.006 for the best LSTM model). What kind of tuning curves do these models have? We found that, for all architecture types, the models also unlearn directional selectivity and have similarly tuned units as for the positional TDT task, with only slightly more velocity-tuned neurons in the intermediate layers (*Figure 6—figure supplement 3A*). Compared with the ART-trained models, the TDT-PV-trained models are tuned more strongly for end-effector position and less strongly for direction (*Figure 6—figure supplement 3B*). As revealed by the decoding analysis, this difference also holds for the distributed representations of direction and position in the networks (*Figure 6—figure supplement 3C*). Further, we found that even when additionally predicting velocity, the TDT task-type models represent velocity less well than ART-trained ones. Next, we looked at the statistics of preferred directions.

## Uniformity and coding invariance

We compared population coding properties to further elucidate the similarity to S1. We measured the distributions of preferred directions and whether coding properties are invariant across different workspaces (reaching planes). Prud'homme and Kalaska found a relatively uniform distribution of preferred directions in primate S1 during a center-out 2D manipulandum-based arm movement task (Figure 7A from *Prud'homme and Kalaska, 1994*). In contrast, most velocity-tuned spindle afferents have preferred directions located along one major axis pointing frontally and slightly away from the body (*Figure 7B*). Qualitatively, it appears that the ART-trained model had more uniformly distributed preferred directions in the middle layers compared to untrained models (*Figure 7C*).

To quantify uniformity, we calculated the total absolute deviation from uniformity in the distribution of preferred directions. The results indicate that the distribution of preferred directions becomes more

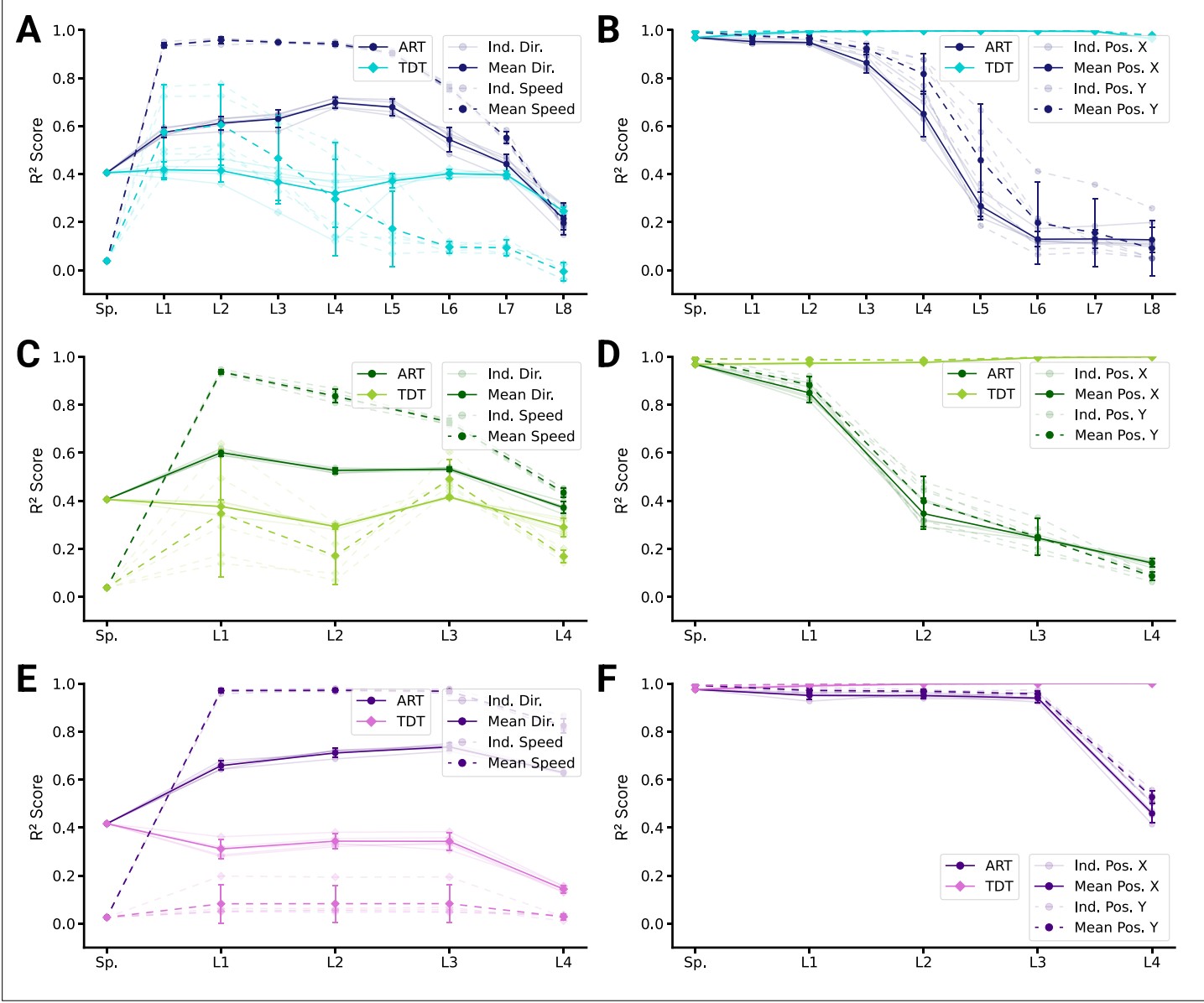

**Figure 6.** Population decoding analysis of action recognition task (ART) vs. trajectory decoding task (TDT) models. (**A**) Population decoding of speed (*light*) and direction (*dark*) for spatial-temporal models for the ART- and TDT-trained models. The faint line shows the $R^2$ score for an individual model; the dark one the mean over all instantiations ($N = 5$). (**B**) Population decoding of end-effector position (*X* and *Y* coordinates) for spatial-temporal models. The faint line shows the $R^2$ score for an individual model; the dark one the mean over all instantiations ($N = 5$). (**C**) Same as (A) but for spatiotemporal models. (**D**) Same as (B) but for spatiotemporal models. (**E**) Same as (A) but for long short-term memory (LSTM) models. (**F**) Same as (B) but for LSTM models.

The online version of this article includes the following figure supplement(s) for figure 6:

**Figure supplement 1.** Analysis of population decoding for action recognition task (ART)-trained and untrained models.

**Figure supplement 2.** Analysis of population decoding for trajectory decoding task (TDT)-trained and untrained models.

**Figure supplement 3.** Results for the position and velocity trajectory decoding task (TDT-PV).

uniform in middle layers for all instantiations of the different model architectures (*Figure 7D*), and that this difference is statistically significant for the spatial-temporal model beginning in layer 3 (layer 3 $t(4) = -3.55$, p=0.024; layer 4 $t(4) = -5.50$, p=0.005; layer 5 $t(4) = -4.60$, p=0.010; layer 6 $t(4) = -6.12$, p=0.004). This analysis revealed that while randomly initialized models also have directionally selective units, those units are less uniformly distributed than in models trained with the ART task. Similar results hold for the spatiotemporal model, for which the difference is statistically significant beginning in layer

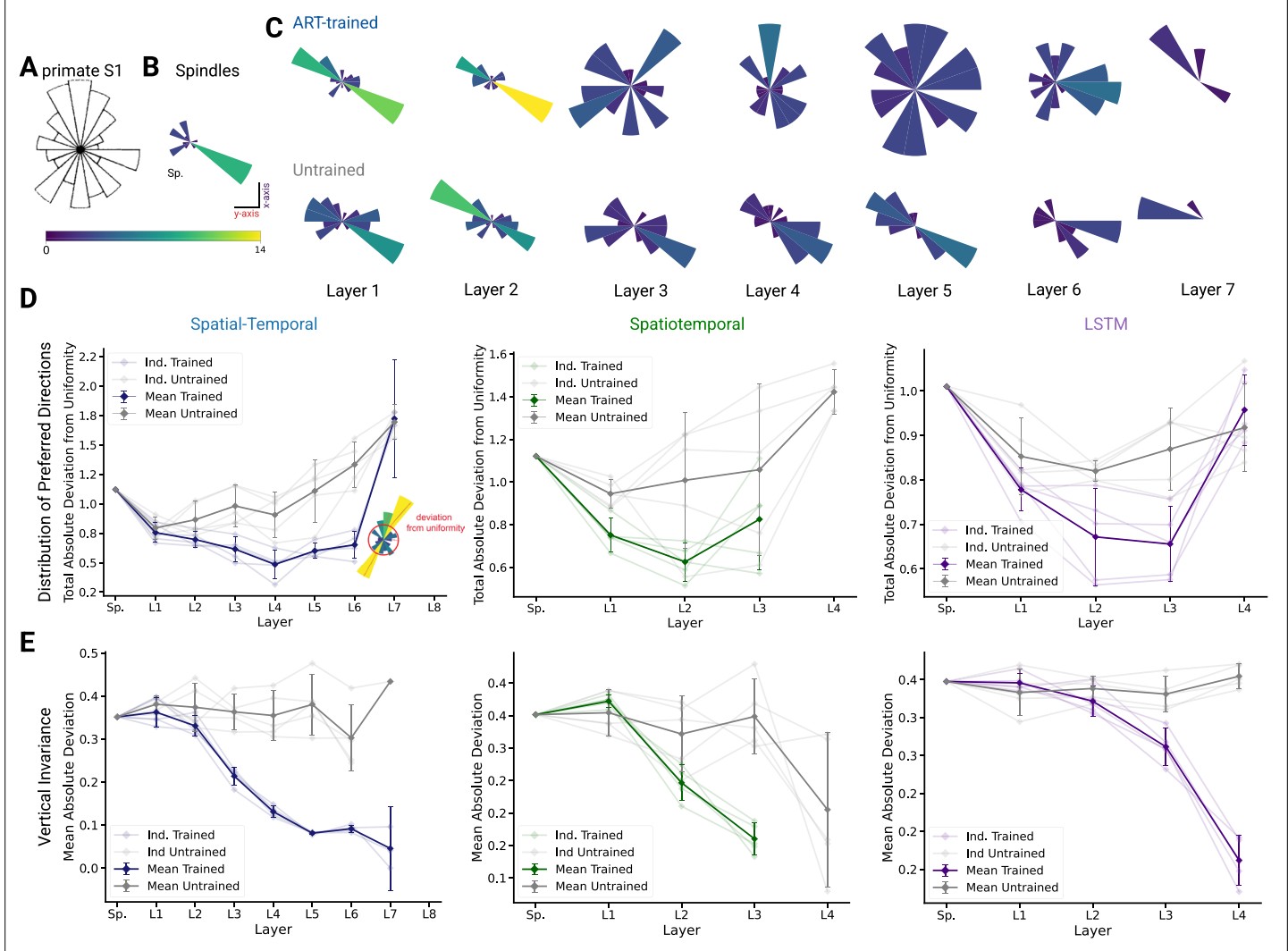

**Figure 7.** Distribution of preferred directions and invariance of representation across workspaces. (**A**) Adopted from **Prud'homme and Kalaska, 1994**; distribution of preferred directions in primate S1. (**B**) Distribution of preferred directions for spindle input. (**C**) Distribution of preferred directions for one spatial-temporal model instantiation (all units with $R^2 > 0.2$ are included). Bottom: the corresponding untrained model. For visibility, all histograms are scaled to the same size and the colors indicate the number of tuned neurons. (**D**) For quantifying uniformity, we calculated the total absolute deviation from the corresponding uniform distribution over the bins in the histogram (red line in inset) for the spatial-temporal model (*left*), the spatiotemporal model (*middle*), and the long short-term memory (LSTM) model (*right*). Normalized absolute deviation from uniform distribution for preferred directions per instantiation is shown ($N = 5$, faint lines) for trained and untrained models as well as mean and 95% confidence intervals over instantiations (solid line; $N = 5$). Note that there is no data for layers 7 and 8 of the trained spatial-temporal model, layer 8 of the untrained spatial-temporal model, and layer 4 of the spatiotemporal model as they have no direction-selective units ($R^2 > 0.2$). (**E**) For quantifying invariance, we calculated mean absolute deviation in preferred orientation for units from the central plane to each other vertical plane (for units with $R^2 > 0.2$). Results are shown for each instantiation ($N = 5$, faint lines) for trained and untrained models plus mean (solid) and 95% confidence intervals over instantiations ($N = 5$). Note that there is no data for layer 4 of the trained spatiotemporal model, as it has no direction-selective units ($R^2 > 0.2$).

The online version of this article includes the following figure supplement(s) for figure 7:

**Figure supplement 1.** Invariance of preferred orientations.

1 (layer 1 $t(4) = -4.25$, p=0.013; layer 2 $t(4) = -2.46$, p=0.070), and for the LSTM beginning in layer 2 (layer 2 $t(4) = -2.88$, p=0.045; layer 3 $t(4) = -3.25$, p=0.031). We also analyzed the preferred directions of the TDT-PV task and found that the distributions deviated from uniform as much, or more, than the untrained distribution (**Figure 6—figure supplement 3D**). Furthermore, positional TDT-trained modules have almost no directionally selective units, lending further support to our hypothesis that

ART models are *more consistent* with Prud'homme and Kalaska's findings (***Prud'homme and Kalaska, 1994***).

Lastly, we tested directly if preferred tuning directions (of tuned units) were maintained across different planes due to the fact that we created trajectories in multiple vertical and horizontal planes. We hypothesized that preferred orientations would be preserved more for trained than untrained models. In order to examine how an individual unit's preferred direction changed across different planes, directional tuning curve models were fit in each horizontal/vertical plane separately (examples in ***Figure 7—figure supplement 1A, B***). To measure representational invariance, we took the mean absolute deviation (MAD) of the preferred tuning direction for directionally tuned units ($R^2 > 0.2$) across planes (see Methods) and averaged over all planes (***Figure 5E***, ***Figure 7—figure supplement 1A***). For the spatial-temporal model across vertical workspaces, layers 3–6 were indeed more invariant in their preferred directions (layer 3: t(4)= –10.30, p=0.0005; layer 4: t(4) = –10.40, p=0.0005; layer 5: t(4) = –10.17, p=0.0005; layer 6: t(4) = –7.37, p=0.0018; ***Figure 5E***; variation in preferred direction illustrated for layer 5 in ***Figure 7—figure supplement 1D*** for trained model and in ***Figure 7—figure supplement 1E*** for untrained). The difference in invariance for the horizontal planes was likewise statistically significant for layers 4–6 (***Figure 7—figure supplement 1A***). A possible reason that the difference in invariance might only become statistically significant one layer later in this setting is that the spindles are already more invariant in the horizontal planes (MAD: $0.225 \pm 2.78e - 17$, mean ± SEM, N=25; ***Figure 7—figure supplement 1B***) than the vertical workspaces (MAD: $0.439 \pm 5e - 17$, mean ± SEM, N=16; ***Figure 7—figure supplement 1C***), meaning that it takes a greater amount of invariance in the trained networks for differences with the untrained networks to become statistically apparent. Across vertical workspaces, the difference in invariance between the ART-trained and untrained models was statistically significant for layers 2–3 for the spatiotemporal model (layer 2 $t(4) = -4.10$, p=0.0149; layer 3 $t(4) = -8.85$, p=0.0009) and for layers 3–4 the LSTM (layer 3 $t(4) = -5.73$, p=0.0046; layer 4 $t(4) = -13.18$, p=0.0002). For these models, the relatively slower increase in invariance in the horizontal direction is exaggerated even more. For the spatiotemporal model, the difference in invariance in the horizontal workspaces becomes statistically significant in layer 3. For the LSTM model, the neuron tuning does not become stronger for the horizontal planes until the recurrent layer (***Figure 7—figure supplement 1A***).

## Discussion
### Task-driven modeling of proprioception

For various anatomical and experimental reasons, recording proprioceptive activity during natural movements is technically challenging (***Delhaye et al., 2018***; ***Kibleur et al., 2020***). Furthermore, 'presenting' particular proprioceptive-only stimuli is difficult, which poses substantial challenges for systems identification approaches. This highlights the importance of developing accurate, normative models that can explain neural representations across the proprioceptive pathway, as has been successfully done in the visual system (***Khaligh-Razavi and Kriegeskorte, 2014***; ***Yamins et al., 2014***; ***Cichy et al., 2016***; ***Yamins and DiCarlo, 2016***; ***Schrimpf et al., 2018***; ***Cadena et al., 2019***; ***Storrs et al., 2021***). To tackle this, we combined human movement data, biomechanical modeling, as well as deep learning to provide a blueprint for studying the proprioceptive pathway.

We presented a task-driven approach to study the proprioceptive system based on our hypothesis that proprioception can be understood normatively, probing two different extremes of learning targets: the trajectory-decoding task encourages the encoding of 'where' information, while the ART focuses on 'what' is done. We created a passive character recognition task for a simulated human biomechanical arm paired with a muscle spindle model and found that deep neural networks can be trained to accurately solve the ART. Inferring the character from passive arm traces was chosen as it is a type of task that humans can easily perform and because it covers a wide range of natural movements of the arm. The perception is also likely fast, so feed-forward processing is a good approximation (while we also find similar results with recurrent models). Additionally, character recognition is an influential task for studying ANNs, for instance MNIST (***Lecun et al., 1998***; ***Illing et al., 2019***). Moreover, when writing movements were imposed onto the ankle with a fixed knee joint, the movement trajectory could be decoded from a few spindles using a population vector model, suggesting that spindle information is accurate enough for decoding (***Albert et al., 2005***). Lastly, while the underlying

movements are natural and of ethological importance for humans, the task itself is only a small subset of human upper-limb function. Thus, it posed an interesting question whether such a task would be *sufficient* to induce representations similar to biological neurons.

We put forth a normative model of the proprioceptive system, which is experimentally testable. This builds on our earlier work (*Sandbrink et al., 2020*), which used a different receptor model (*Prochazka and Gorassini, 1998a*). Here, we also include positional sensing and an additional task to directly test our hypothesis of action coding vs. the canonical view of proprioception (trajectory decoding). We also confirm (for different receptor models) that in ART-trained models, but not in untrained models, the intermediate representations contain directionally selective neurons that are uniformly distributed (*Sandbrink et al., 2020*). Furthermore, we had predicted that earlier layers, and in particular muscle spindles, have a biased, bidirectionally tuned distribution. This distribution was later experimentally validated for single units in the cuneate nucleus (*Versteeg et al., 2021*). Here, we still robustly find this result but with different spindle models (*Dimitriou and Edin, 2008b*). However, interestingly, these PD distribution results do not hold when neural networks with identical architectures are trained on trajectory decoding. In those models, directionally tuned neurons do not emerge but are 'unlearned' in comparison to untrained models for TDT (*Figure 5—figure supplement 3*).

The distribution of preferred directions becomes more uniform over the course of the processing hierarchy for the ART-trained models, similar to the distribution of preferred tuning in somatosensory cortex (*Figure 5A–D*, *Prud'homme and Kalaska, 1994*). This does not occur either in the untrained (*Figure 5C–D*) or the PV-TDT models (*Figure 4*, *Figure 5—figure supplement 3*), which instead maintain an input distribution centered on the primary axis of preferred directions of the muscular tuning curves. Furthermore, the ART-trained models make a prediction about the distribution of preferred directions along the proprioceptive pathway. For instance, we predict that in the brainstem – that is cuneate nucleus – preferred directions are aligned along major axes inherited from muscle spindles that correspond to biomechanical constraints (consistent with *Versteeg et al., 2021*). A key element of robust object recognition is invariance to task-irrelevant variables (*Yamins and DiCarlo, 2016*; *Serre, 2019*). In our computational study, we could probe many different workspaces (26 horizontal and 18 vertical) to reveal that training on the character recognition task makes directional tuning more invariant (*Figure 5E*). This, together with our observation that directional tuning is simply inherited from muscle spindles, highlights the importance of sampling the movement space well, as also emphasized by pioneering experimental studies (*Jackson et al., 2007*). We also note that the predictions depend on the musculoskelatal model and the movement statistics. In fact, we predict that, for example, distributions of tuning and invariances might be different in mice, a species that has a different body orientation from primates.

## Limitations and future directions

Using task-driven modeling we could show that tuning properties consistent with known biological tuning curve properties emerged from models trained on a higher-order goal, namely action recognition. This has also been shown in other sensory systems, where models were typically trained on complex, higher-order tasks (*Khaligh-Razavi and Kriegeskorte, 2014*; *Yamins et al., 2014*; *Cichy et al., 2016*; *Yamins and DiCarlo, 2016*; *Schrimpf et al., 2018*; *Cadena et al., 2019*; *Storrs et al., 2021*). While the ART task likely captures aspects of higher-order tasks that would be needed to predict the 'where' and the 'what' (to use the visual system analogy), it is not exhaustive, and it is likely that proprioception multiplexes goals, such as postural representation and higher-order tasks (like action recognition). Both the 'what' and the 'where' are important for many other behaviors (such as in state estimation for motor control). Therefore, future work can design new tasks that make experimentally testable predictions for coding in the proprioceptive pathway.

Our model only encompasses proprioception and was trained in a supervised fashion. However, it is quite natural to interpret the supervised feedback stemming from other senses. For instance, the visual system could naturally provide information about hand localization or about the type of character. The motor system could also provide this information during voluntary movement. Thus, one future direction should be multi-modal integration not only from the motor system, but from critical systems like vision.

We used different types of temporal convolutional and recurrent network architectures. In future work, it will be important to investigate emerging, perhaps more biologically relevant architectures to

better understand how muscle spindles are integrated in upstream circuits. While we used spindle Ia and II models, it is known that multiple receptors, namely cutaneous, joint, and muscle receptors, play a role for limb localization and kinesthesia (*Gandevia et al., 2002*; *Mileusnic et al., 2006*; *Aimonetti et al., 2007*; *Blum et al., 2017*; *Delhaye et al., 2018*). For instance, a recent simulation study by *Kibleur et al., 2020* highlighted the complex spatiotemporal structure of proprioceptive information at the level of the cervical spinal cord. Furthermore, due to fusimotor drive receptor activity can be modulated by other modalities, for example, vision (*Ackerley et al., 2019*). In the future, models for other afferents, Golgi tendon organ including the fusimotor drive as well as cutaneous receptors can be added to study their role in the context of various tasks (*Hausmann et al., 2021*). Furthermore, as highlighted in the introduction, we studied proprioception as an open-loop system. Future work should study the effect of active motor control on proprioception.

## Methods

### Proprioceptive character trajectories: dataset and tasks

The character trajectories dataset

The movement data for our task was obtained from the UCI Machine Learning Repository character trajectories dataset (*Williams, 2008*; *Williams et al., 2006*). In brief, the dataset contains 2858 pen-tip trajectories for 20 single-stroke characters (excluding f, i, j, k, t, and x, which were multi-stroke in this dataset) in the Latin alphabet, written by a single person on an Intuos 3 Wacom digitization tablet providing pen-tip position and pressure information at 200 Hz. The size of the characters was such that they all approximately fit within a $1 \times 1$ cm$^2$ grid. Since we aimed to study the proprioception of the whole arm, we first interpolated the trajectories to lie within a $10 \times 10$ cm$^2$ grid and discarded the pen-tip pressure information. Trajectories were interpolated linearly while maintaining the velocity profiles of the original trajectories. Empirically, we found that on average it takes three times longer to write a character in the $10 \times 10$ cm$^2$ grid than in the small $1 \times 1$ one. Therefore, the time interval between samples was increased from 5 ms (200 Hz) to 15 ms (66.7 Hz) when interpolating trajectories. The resulting 2858 character trajectories served as the basis for our end-effector trajectories.

Computing joint angles and muscle length trajectories

Using these end-effector trajectories, we sought to generate realistic proprioceptive inputs for passively executed movements. For this purpose, we used an open-source musculoskeletal model of the human upper limb, the upper extremity dynamic model by *Saul et al., 2015*; *Holzbaur et al., 2005*. The model includes 50 Hill-type muscle-tendon actuators crossing the shoulder, elbow, forearm, and wrist. While the kinematic foundations of the model enable it with 15 DoFs, 8 DoFs were eliminated by enforcing the hand to form a grip posture. We further eliminated 3 DoFs by disabling the model to have elbow rotation, wrist flexion, and rotation. The four remaining DoFs are elbow flexion ($\theta_{ef}$), shoulder rotation ($\theta_{sr}$), shoulder elevation, that is, thoracohumeral angle ($\theta_{se}$) and elevation plane of the shoulder ($\theta_{sep}$).

The first step in extracting the spindle activations involved computing the joint angles for the 4 DoFs from the end-effector trajectories using constrained inverse kinematics. We built a 2-link 4 DoF arm with arm-lengths corresponding to those of the upper extremity dynamic model (*Holzbaur et al., 2005*). To determine the joint-angle trajectories, we first define the forward kinematics equations that convert a given joint-angle configuration of the arm to its end-effector position. For a given joint-angle configuration of the arm $\mathbf{q} = [\theta_{ef}, \theta_{sr}, \theta_{se}, \theta_{sep}]^T$, the end-effector position $\mathbf{e} \in \mathbb{R}^3$ in an absolute frame of reference $\{S\}$ centered on the shoulder is given by

$$\mathbf{e} = R_S(R_L \mathbf{e}_0 + \mathbf{l}_0) =: F(\mathbf{q}), \tag{1}$$

with position of the end-effector (hand) $\mathbf{e}_0$ and elbow $\mathbf{l_0}$ when the arm is at rest and rotation matrices

$$R_S \quad = R_Y(\theta_{se})R_Z(\theta_{sep})R_Y(-\theta_{se})R_Y(\theta_{sr}), \tag{2}$$

$$R_L = R_X(\theta_{ef}). \tag{3}$$

Thereby, $R_S$ is the rotation matrix at the shoulder joint, $R_L$ is the rotation matrix at the elbow obtained by combinations of intrinsic rotations around the X, Y, and Z axes which are defined according to the upper extremity dynamic model (*Holzbaur et al., 2005*), treating the joint angles as Euler angles and $R_X, R_Y, R_Z$ – the three basic rotation matrices.

Given the forward kinematics equations, the joint angles **q** for an end-effector position **e** can be obtained by iteratively solving a constrained inverse kinematics problem for all times $t = 0 \ldots T$:

$$\text{minimize } \|\mathbf{q}(t) - \mathbf{q}(t - 1)\|$$
$$\text{subject to } \|F(\mathbf{q}(t)) - \mathbf{e}(t)\| = 0, \tag{4}$$
$$\theta_{min} \leq \theta \leq \theta_{max} \quad \forall \, \theta \, \in \, \{\theta_{ef}, \theta_{sr}, \theta_{se}, \theta_{sep}\},$$

where $\mathbf{q}(-1)$ is a natural pose in the center of the workspace (see *Figure 2D*) and each $\mathbf{q}(t)$ is a posture pointing to $\mathbf{e}(t)$, while being close to the previous posture $\mathbf{q}(t - 1)$. Thereby, $\{\theta_{min}, \theta_{max}\}$ define the limits for each joint angle. For a given end-effector trajectory $\mathbf{e}(t)$, joint-angle trajectories are thus computed from the previous time point in order to generate smooth movements in joint space. This approach is inspired by *D'Souza et al., 2001*.

Finally, for a given joint trajectory $\mathbf{q}(t)$, we passively moved the arm through the joint-angle trajectories in the OpenSim 3.3 simulation environment (*Delp et al., 2007*; *Seth et al., 2011*), computing at each time point the equilibrium muscle lengths $\mathbf{m}(t) \in \mathbb{R}^{25}$, since the actuation of the 4 DoFs is achieved by 25 muscles. For simplicity, we computed equilibrium muscle configurations given joint angles as an approximation to passive movement.

## Proprioceptive inputs

While several mechanoreceptors provide proprioceptive information, including joint receptors, Golgi tendon organs and skin stretch receptors, the muscle spindles are regarded as the most important for conveying position and movement-related information (*Macefield and Knellwolf, 2018*; *Proske and Gandevia, 2012*; *Prochazka and Gorassini, 1998a*; *Prochazka and Gorassini, 1998b*). Here, we are inspired by Dimitriou and Edin's recordings from human spindles (*Dimitriou and Edin, 2008a*). They found that both Ia and II units are well predicted by combinations (for parameters $k_1 \ldots k_5$) of muscle length $l$, muscle velocity $l'$, acceleration $l''$, and EMG:

$$k1 + k2 \cdot l + k3 \cdot l' + k4 \cdot l'' + k5 \cdot EMG \tag{5}$$

As we model passive movement, the associated EMG activity is negligible. To simplify the aggregate information flowing from one muscle (via multiple Ia and II spindles), we consider a more generic/functional representation of proprioceptive information as consisting of muscle length and velocity signals, which are approximately conveyed by muscle spindles during passive movements. Therefore, in addition to the equilibrium muscle lengths $\mathbf{m}(t)$, we input the muscle velocity $\mathbf{v}(t)$ signal obtained by taking the first derivative. Taken together, $\{\mathbf{m}(t), \mathbf{v}(t)\}$ form the proprioceptive inputs that we use to train models of the proprioceptive system.

## A scalable proprioceptive character trajectories dataset

We move our arms in various configurations and write at varying speeds. Thus, several axes of variation were added to each (original) trajectory by (1) applying affine transformations such as scaling, rotation, and shear, (2) modifying the speed at which the character is written, (3) writing the character at several locations (chosen from a grid of candidate starting points) in the 3D workspace of the arm, and (4) writing the characters on either transverse (horizontal) or frontal (vertical) planes, of which there were 26 and 18, respectively, placed at a spatial distance of 3 cm from each other (see *Table 1* for parameter ranges). We first generated a dataset of end-effector trajectories of 1 million samples by generating variants of each original trajectory, by scaling, rotating, shearing, translating, and varying its speed. For each end-effector trajectory, we compute the joint-angle trajectory by performing inverse kinematics. Subsequently, we simulate the muscle length and velocity trajectories. Since different characters take different amounts of time to be written, we pad the movements with static postures corresponding to the starting and ending postures of the movement, and jitter the beginning of the writing to maintain ambiguity about when the writing begins.

**Table 1.** Variable range for the data augmentation applied to the original pen-tip trajectory dataset.

Furthermore, the character trajectories are translated to start at various starting points throughout the arm's workspace, overall yielding movements in 26 horizontal and 18 vertical planes.

| Type of variation | Levels of variation |
|---|---|
| Scaling | [0.7×, 1×, 1.3×] |
| Rotation | [-$\pi$/6, -$\pi$/12, 0, $\pi$/12, $\pi$/6] |
| Shearing | [-$\pi$/6, -$\pi$/12, 0, $\pi$/12, $\pi$/6] |
| Translation | Grid with a spacing of 3 cm |
| Speed | [0.8×, 1×, 1.2×, 1.4×] |
| Plane of writing | [Horizontal (26), Vertical (18)] |

From this dataset of trajectories, we selected a subset of trajectories such that the integral of joint-space jerk (third derivative of movement) was less than 1 rad/s$^3$ to ensure that the arm movement is sufficiently smooth. Among these, we picked the trajectories for which the integral of muscle-space jerk was minimal, while making sure that the dataset was balanced in terms of the number of examples per class, resulting in 200,000 samples. The final dataset consists of muscle length and velocity trajectories from each of the 25 muscles over a period of 320 time points, simulated at 66.7 Hz (i.e., 4.8 s). In other words, the dimensionality of the proprioceptive inputs in our tasks is 25 × 320 × 2. The dataset was then split into a training, validation, and test set with a 72-8-20 ratio.

## ART and TDT

Having simulated a large scale dataset of proprioceptive character trajectories, we designed two tasks: (1) the ART to classify the identity of the character based on the proprioceptive inputs, and (2) the TDT to decode the end-effector coordinates (at each time step), from proprioceptive inputs. Baseline models (SVMs for the ART and linear regression for TDT) were first trained to investigate the difficulty of the task, followed by a suite of deep neural networks that aim to model the proprioceptive pathway.

As a control, we also implemented the trajectory decoding task (TDT-PV) to decode both the end-effector coordinates and velocity (at each time step) from proprioceptive inputs.

## Low-dimensional embedding of population activity

To visualize population activity (of kinematic or network representations), we created low-dimensional embeddings of the proprioceptive inputs as well as the internal representations the neural network models, along time, and space/muscles dimensions. To this end, we first used principal components analysis to reduce the space to 50 dimensions, typically retaining around 75–80% of the variance. We then used t-SNE (*Maaten and Hinton, 2008*) using sklearn (*Pedregosa et al., 2011*) with a perplexity of 40 for 300 iterations, to reduce these 50 dimensions down to 2 for visualization.

## SVM analysis for action recognition

To establish a baseline performance for multi-class recognition, we used pairwise SVMs with the one-against-one method (*Hsu and Lin, 2002*). That is, we train $\binom{20}{2}$ pairwise (linear) SVM classifiers (*Figure 3C*) and at test time implement a voting strategy based on the confidences of each classifier to determine the class identity. We trained SVMs for each input modality (end-effector trajectories, joint-angle trajectories, muscle fiber-length trajectories, and proprioceptive inputs) to determine how the format affects performance. All pairwise classifiers were trained using a hinge loss, and cross-validation was performed with nine regularization constants spaced logarithmically between $10^{-4}$ and $10^4$.

## Baseline linear regression model for trajectory decoding

To establish how well one could decode end-effector coordinates from the joint, muscle, and proprioceptive inputs, we trained linear regressors with ordinary least-squares loss using stochastic gradient descent until the validation loss saturated (with a tolerance of $10^{-3}$). Inputs and outputs to the model were first transformed using a standard scalar to center (remove the mean) and scale to unit variance over each feature in order to train faster. At test time, the same scalars were reused. Decoding error

was determined as the squared error (L-2 norm) of the predicted and true end-effector coordinates in 3D.

## Models of the proprioceptive system

We trained two types of convolutional networks and one type of recurrent network on the two tasks. Each model is characterized by the layers used – convolutional and/or recurrent – which specify how the spatial and temporal information in the proprioceptive inputs is processed and integrated.

Each convolutional layer contains a set of convolutional filters of a given kernel size and stride, along with response normalization and a point-wise non-linearity. The convolutional filters can either be 1D, processing only spatial or temporal information, or 2D, processing both types of information simultaneously. For response normalization we use layer normalization (*Ba et al., 2016*), a commonly used normalization scheme to train deep neural networks, where the response of a neuron is normalized by the response of all neurons of that layer. As point-wise non-linearity, we use rectified linear units. Each recurrent layer contains a single LSTM cell with a given number of units that process the input one time step at a time.

Depending on what type of convolutional layers are used and how they are arranged, we classify convolutional models into two subtypes: (1) spatial-temporal and (2) spatiotemporal networks. Spatial-temporal networks are formed by combining multiple 1D spatial and temporal convolutional layers. That is, the proprioceptive inputs from different muscles are first combined to attain a condensed representation of the 'spatial' information in the inputs, through a hierarchy of spatial convolutional layers. This hierarchical arrangement of the layers leads to increasingly larger receptive fields in spatial (or temporal) dimension that typically (for most parameters) gives rise to a representation of the whole arm at some point in the hierarchy. The temporal information is then integrated using temporal convolutional layers. In the spatiotemporal networks, multiple 2D convolutional layers where convolutional filters are applied simultaneously across spatial and temporal dimensions are stacked together. The LSTM models on the other hand are formed by combining multiple 1D spatial convolutional layers and a single LSTM layer at the end of a stack of spatial filters that recurrently processes the temporal information. For each network, the features at the final layer are mapped by a single fully connected layer onto either a 20D (logits) or a 3D output (end-effector coordinates).

---

**Table 2.** Hyper-parameters for neural network architecture search.

To form candidate networks, first a number of layers (per type) is chosen, ranging from 2 to 8 (in multiples of 2) for spatial-temporal models and 1–4 for the spatiotemporal and long short-term memory (LSTM) ones. Next, a spatial and temporal kernel size per Layer is picked where relevant, which remains unchanged throughout the network. For the spatiotemporal model, the kernel size is equal in both the spatial and temporal directions in each layer. Then, for each layer, an associated number of kernels/feature maps is chosen such that it never decreases along the hierarchy. Finally, a spatial and temporal stride is chosen. For the LSTM networks, the number of recurrent units is also chosen. All parameters are randomized independently and 50 models are sampled per network type. Columns 2–4: Hyper-parameter values for the top-performing models in the ART. The values given under the spatial rows count for both the spatial and temporal directions for the spatiotemporal model.

|  | Hyper-parameters | Spatial-temporal | Spatiotemporal | LSTM |
|---|---|---|---|---|
| Num. layers | [1, 2, 3, 4] | 4+4 | 4 | 3+1 |
| Spatial kernels (pL) | [8, 16, 32, 64] | [8,16,16,32] | [8, 8, 32, 64] | [8, 16, 16] |
| Temporal kernels (pL) | [8, 16, 32, 64] | [32, 32, 64, 64] | n/a | n/a |
| Spatial kernel size | [3, 5, 7, 9] | 7 | 7 | 3 |
| Temporal kernel size | [3, 5, 7, 9] | 2 | n/a | n/a |
| Spatial stride | [1, 2] | 9 | 2 | 1 |
| Temporal stride | [1, 2, 3] | 3 | n/a | n/a |
| Num. recurrent units | [128, 256] | n/a | n/a | 256 |

For each specific network type, we experimented with the following hyper-parameters: number of layers, number and size of spatial and temporal filters, and their corresponding stride (see *Table 2*). Using this set of architectural parameters, 50 models of each type were randomly generated. Notably, we trained the same model (as specified by the architecture) on both the ART and the TDT position tasks.

## Network training and evaluation procedure

The action-recognition trained models were trained by minimizing the softmax cross entropy loss using the Adam Optimizer (*Kingma and Ba, 2014*) with an initial learning rate of 0.0005, batch size of 256, and decay parameters ($\beta_1$ and $\beta_2$) of 0.9 and 0.999. During training, we monitored performance on the held-out validation set. When the validation error did not improve for five consecutive epochs, we decreased the learning rate by a factor of 4. After the second time the validation error saturated, we ended the training and evaluated accuracy of the networks on the test set. Overall, we observe that the trained networks generalized well to the test data, even though the shallower networks tended to overfit (see *Figure 3—figure supplement 1A*).

The TDT-trained models, on the other hand, were trained to minimize the mean squared error between predicted and true trajectories. Hyper-parameter settings for the optimizer, batch size, and early stopping procedure used during training remained the same across both tasks. Here, we observe that train and test decoding errors were highly correlated, and thereby achieve excellent generalization to test data.

Note that only the best five instantiations of all three model types were trained on the TDT-PV task. Here, the trajectories consisted of concatenated position and velocity targets. Since the position and velocity magnitudes differed, each target dimension was scaled by their range on the training set (min-max scaling) to ensure the trained models decoded both variables similarly.

## Comparison with untrained models

For each of the three types of models, the architecture belonging to the best performing model on the ART (as identified via the hyper-parameter search) was chosen as the basis of the analysis (*Table 2*). The resulting sizes of each layer's representation across the hierarchy are given in *Table 3*. For each different model type, five sets of random weights were initialized and saved. Then, each instantiation was trained on both ART and TDT using the same training procedure as described in the previous section, and the weights were saved again after training. This gives a before and after structure for each run that allows us to isolate the effect of task training.

**Table 3.** Size of representation at each layer for best-performing architecture of each network type (spatial × temporal × filter dimensions).

| Layer | Dimension | Layer | Dimension | Layer | Dimension |
|---|---|---|---|---|---|
| Input | 25 × 320 × 2 | Input | 25 × 320 × 2 | Input | 25 × 320 × 2 |
| SC0 | 13 × 320 × 8 | STC0 | 13 × 160 × 8 | SC0 | 25 × 320 × 8 |
| SC1 | 7 × 320 × 16 | STC1 | 7 × 80 × 8 | SC1 | 25 × 320 × 16 |
| SC2 | 4 × 320 × 16 | STC2 | 4 × 40 × 32 | SC2 | 25 × 320 × 16 |
| SC3 | 2 × 320 × 32 | STC3 | 2 × 20 × 64 | R | 256 × 320 |
| TC0 | 2 × 107 × 32 | | | | |
| TC1 | 2 × 36 × 32 | | | | |
| TC2 | 2 × 12 × 64 | | | | |
| TC3 | 2 × 4 × 64 | | | | |

## Population comparisons

### Centered kernel alignment

In order to provide a population-level comparison between the trained and untrained models (*Figure 4A*), we used linear CKA for a high-level comparison of each layer's activation patterns (*Kornblith et al., 2019*). CKA is an alternative that extends canonical correlation analysis (CCA) by weighting activation patterns by the eigenvalues of the corresponding eigenvectors (*Kornblith et al., 2019*). As such, it maintains CCA's invariance to orthogonal transformations and isotropic scaling, yet retains a greater sensitivity to similarities. Using this analysis, we quantified the similarity of the activation of each layer of the trained models with those of the respective untrained models in response to identical stimuli comprising 50% of the test set for each of the five model instantiations.

### Representational similarity analysis

Representational similarity analysis (RSA) is a tool to investigate population level representations among competing models (*Kriegeskorte et al., 2008*). The basic building block of RSA is a RDM. Given stimuli $\{s_1, s_2, \ldots, s_n\}$ and vectors of population responses $\{r_1, r_2, \ldots, r_n\}$, the RDM is defined as:

$$\text{RDM}_{ij} = 1 - \frac{\text{cov}(r_i, r_j)}{\sqrt{\text{var}(r_i) \cdot \text{var}(r_j)}}. \tag{6}$$

One of the main advantages of RDMs is that they characterize the geometry of stimulus representation in a way that is independent of the dimensionality of the feature representations, so we can easily compare between arbitrary representations of a given stimulus set. Example RDMs for proprioceptive inputs as well as the final layer before the readout for the best models of each type are shown in *Figure 4C*, *Figure 4—figure supplement 1B*. Each RDM is computed for a random sample of 4000 character trajectories (200 from each class) by using the correlation distance between corresponding feature representations. To compactly summarize how well a network disentangles the stimuli we compare the RDM of each layer to the RDM of the ideal observer, which has an RDM with perfect block structure (with dissimilarity values 0 for all stimuli of the same class and 1 (100 percentile) otherwise; see *Figure 3—figure supplement 1B*).

## Single unit analysis

### Comparing the tuning curves

To elucidate the emerging coding properties of single units, we determined label specificity and fit tuning curves. Specifically, we focused on kinematic properties such as direction, velocity, acceleration, and position of the end-effector (*Figure 5* and *Figure 7*). For computational tractability, 20,000 of the original trajectories were randomly selected for the spatial-temporal and spatiotemporal models, and 10,000 for the LSTM models. In convolutional layers in which the hidden layers had a reduced temporal dimensionality, the input trajectory was downsampled. Only those time points were kept that correspond to the center of the receptive fields of the units in the hidden layers.

A train-test split of 80-20 was used, split between trajectories (so that a single trajectory was only used for training or only used for testing, but never both, eliminating the possibility of correlations between train and test resulting from temporal network dynamics). The tuning curves were fit and tested jointly on all movements in planes with a common orientation, vertical or horizontal. The analysis was repeated for each of the five trained and untrained models. For each of the five different types of tuning curves (the four biological ones and label specificity) and for each model instantiation, distributions of test scores were computed.

When plotting comparisons between different types of models (ART, TDT, and untrained), the confidence interval for the mean (CLM) using an $\alpha = 5\%$ significance level based on the t-statistic was displayed.

### Label tuning (selectivity index)

The networks' ability to solve the proprioceptive task poses the question if individual units serve as character detectors. To this end, SVMs were fit with linear kernels using a one-vs.-rest strategy for multi-class classification based on the firing rate of each unit, resulting in linear decision boundaries for each letter. Each individual SVM classifies whether the trajectory belongs to a certain character or not,

based on that unit's firing rates. For each SVM, auROC was calculated, giving a measure of how well the label can be determined based on the firing rate of an individual unit alone. The label specificity of that unit was then determined by taking the maximum over all characters. Finally, the auROC score was normalized into a selectivity index: 2((auROC)–0.5).

## Position, direction, velocity, and acceleration

For the kinematic tuning curves, the coefficient of determination $R^2$ on the test set was used as the primary metric of evaluation. These tuning curves were fit using ordinary least squares linear regression, with regularization proving unnecessary due to the high number of data points and the low number of parameters (2-3) in the models.

## Position tuning

Position $\begin{pmatrix} x \\ y \end{pmatrix}$ is initially defined with respect to the center of the workspace. For trajectories in a *horizontal* plane (workspace), a position vector was defined with respect to the starting position $\begin{pmatrix} x_0 \\ y_0 \end{pmatrix}$ of each trace, $\vec{\rho_t} = \begin{pmatrix} x - x_0 \\ y - y_0 \end{pmatrix}$. This was also represented in polar coordinates $\vec{\rho_t} = \begin{pmatrix} \rho_t \\ \phi_t \end{pmatrix}$, where $\phi_t \in (-\pi, \pi]$ is the angle measured with the counterclockwise direction defined as positive between the position vector and the vector $\begin{pmatrix} 1 \\ 0 \end{pmatrix}$, that is the vector extending away from the body, and $\rho_t = \|\vec{\rho_t}\| = \sqrt{(x - x_0)^2 + (y - y_0)^2}$. Positional tuning of the neural activity $N$ of node $\nu$ was evaluated by fitting models both using Cartesian coordinates,

$$N_\nu(\vec{\rho_t}) = \alpha_1 \, x_t + \alpha_2 \, y_t + \beta \tag{7}$$

as well as polar ones,

$$N_\nu(\vec{\rho_t}) = \alpha \, \rho_t \cos(\phi_t - \phi_{\mathrm{PD}}) + \beta \tag{8}$$

where $\phi_{\mathrm{PD}}$ is a parameter representing a neuron's preferred direction for position. For trajectories in the *vertical* plane, all definitions are equivalent, but with coordinates $(y, z)^T$.

## Direction

In order to examine the strength of kinematic tuning, tuning curves relating direction, velocity, and acceleration to neural activity were fitted. Since all trajectories take place either in a horizontal or vertical plane, the instantaneous velocity vector at time $t$ can be described in two components as $\vec{v_t} = \begin{pmatrix} \dot{x_t} \\ \dot{y_t} \end{pmatrix}$, or $(y, z)^T$ for trajectories in a vertical plane, or alternately in polar coordinates, $\vec{v_t} = \begin{pmatrix} s_t \\ \theta_t \end{pmatrix}$, with $\theta_t \in (-\pi, \pi]$ representing the angle between the velocity vector and the x-axis, and $s_t = \|\vec{v_t}\| = \sqrt{x^2 + y^2}$ representing the speed.

First, a tuning curve was fit that excludes the magnitude of velocity but focuses on the instantaneous direction, putting the angle of the polar representation of velocity $\theta_t$ in relation to each neuron's preferred direction $\theta_{\mathrm{PD}}$.

$$N_\nu(\theta_t) = \alpha \cos(\theta_t - \theta_{\mathrm{PD}}) + \beta \tag{9}$$

To fit this model, *Equation 9* was re-expressed as a simple linear sum using the cosine sum and difference formula $\cos(\alpha + \beta) = \cos\alpha\cos\beta - \sin\alpha\sin\beta$, a reformulation that eases the computational burden of the analysis significantly (*Georgopoulos et al., 1982*). In this formulation, the equation for directional tuning becomes:

$$N_\nu(\theta_t) = \alpha_1 \cos\theta_t + \alpha_2 \sin\theta_t + \beta \tag{10}$$

The preferred direction $\theta_{\text{PD}}$ is now contained in the coefficients $\alpha_1 = \alpha \cos \theta_{\text{PD}}$ and $\alpha_2 = \alpha \sin \theta_{\text{PD}}$.

The quality of fit of this type of tuning curve was visualized using polar scatter plots in which the angle of the data point corresponds to the angle $\theta$ in the polar representation of velocity and the radius corresponds to the node's activation. In the figures the direction of movement was defined so that 0° ($Y$) corresponds to movement to the right of the body and progressing counterclockwise, a movement straight (forward) away from the body corresponds to 90° ($X$) (**Figure 5**; **Figure 7**).

### Speed

Two linear models for activity $N$ at a node $\nu$ for velocity were fit.

The first is based on its magnitude, speed,

$$N_\nu(\vec{v}_t) = \alpha \, s_t + \beta \tag{11}$$

### Velocity

The second velocity-based tuning curve factors in both directional and speed components:

$$N_\nu(\vec{v}_t) = \alpha \, s_t \, \cos(\theta_t - \theta_{\text{PD}}) + \beta \tag{12}$$

The quality of fit of this type of tuning curve was visualized using polar filled contour plots in which the angle of the data point corresponds to the angle $\theta$ in the polar representation of velocity, the radius corresponds to the speed, and the node's activation is represented by the height. For the visualizations (**Figure 5B**), to cover the whole range of angle and radius given a finite number of samples, the activation was first linearly interpolated. Then, missing regions were filled in using nearest neighbor interpolation. Finally, the contour was smoothed using a Gaussian filter.

### Acceleration

Acceleration is defined analogously to velocity by $\vec{a}_t = \begin{pmatrix} \ddot{x} \\ \ddot{y} \end{pmatrix}$ and $a_t = \|\vec{a}_t\| = \sqrt{\ddot{x}^2 + \ddot{y}^2}$. A simple linear relationship with acceleration magnitude was tested:

$$N_\nu(\vec{a}_t) = \alpha \, a_t + \beta \tag{13}$$

In subsequent analyses, scores were excluded if they were equal to 1 (indicating a dead neuron whose output was constant) or if they were less than –0.1 (indicating a fit that did not converge).

### Classification of neurons into different types

The neurons were classified as belonging to a certain type if the corresponding kinematic model yielded a test $R^2 > 0.2$. Seven different model types were evaluated:

1. Direction tuning
2. Velocity tuning
3. Direction and velocity tuning
4. Position (Cartesian)
5. Position (polar)
6. Acceleration tuning
7. Label specificity

These were treated as distinct classes for the purposes of classification.

## Population decoding analysis

We also performed population-level decoding analysis for the kinematic tuning curve types. The same datasets were used as for the single-cell encoding analysis, except with switched predictors and targets. The firing rates of all neurons in a hidden layer at a single time point were jointly used as predictors for the kinematic variable at the center of the receptive field at the corresponding input layer.

This analysis was repeated for each of the following kinematic variables:

1. Direction

2. Speed
3. *X* position (Cartesian)
4. *Y* position (Cartesian)

For each of these, the accuracy was evaluated using $R^2$ score. The encoding strength for *X* and *Y* position in Cartesian coordinates was additionally jointly evaluated by calculating the average distance between true and predicted points of the trajectory. To prevent over-fitting, ridge regularization was used with a regularization strength of 1.

## Distribution of preferred directions

Higher-order features of the models were also evaluated and compared between the trained models and their untrained counterparts. The first property was the distribution of preferred directions fit for all horizontal planes in each layer. If a neuron's direction-only tuning yields a test $R^2 > 0.2$, its preferred direction was included in the distribution. Within a layer, the preferred directions of all neurons were binned into 18 equidistant intervals in order to enable a direct comparison with the findings by *Prud'homme and Kalaska, 1994*. They found that the preferred directions of tuning curves were relatively evenly spread in S1; our analysis showed that this was not the case for muscle spindles. Thus, we formed the hypothesis that the preferred directions in the trained networks were more uniform in the trained networks than in the random ones. For quantification, absolute deviation from uniformity was used as a metric. To calculate this metric, the deviation from the mean height of a bin in the circular histograms was calculated for each angular bin. Then, the absolute value of this deviation was summed over all bins. We then normalize the result by the number of significantly directionally tuned neurons in a layer, and compare the result for the trained and untrained networks.

## Preferred direction invariance

We also hypothesized that the representation in the trained network would be more invariant across different horizontal and vertical planes, respectively. To test this, directional tuning curves were fit for each individual plane. A central plane was chosen as a basis of comparison (plane at $z = 0$ for the horizontal planes and at $x = 30$ for vertical). Changes in preferred direction of neurons are shown for spindles, as well as for neurons of layer 5 of one instantiation of the trained and untrained spatial-temporal model. Generalization was then evaluated as follows: for neurons with $R^2 > 0.2$, the average deviation of the neurons' preferred directions over all different planes from those in the central plane was summed up and normalized by the number of planes and neurons, yielding a total measure for the neurons' consistency in preferred direction in any given layer. If a plane had fewer than three directionally tuned neurons, its results were excluded.

## Statistical testing

To test whether differences were statistically significant between trained and untrained models, paired t-tests were used with a pre-set significance level of $\alpha = 0.05$.

## Software

We used the scientific Python3 stack (python.org): Numpy, Pandas, Matplotlib, SciPy (*Virtanen et al., 2020*), and scikit-learn (*Pedregosa et al., 2011*). OpenSim (*Delp et al., 2007*; *Seth et al., 2011*; *Saul et al., 2015*) was used for biomechanics simulations and Tensorflow was used for constructing and training the neural network models (*Abadi et al., 2016*).

## Code and data

Code and data is available at https://github.com/amathislab/DeepDraw, (*Sandbrink et al., 2023* copy archived at swh:1:rev:5785af7b25375e58c1d26a7ccd1787596474287f).

## Acknowledgements

We are grateful to the Mathis Lab, Mathis Group, and Bethge Group for comments on earlier versions of the manuscript, and Travis DeWolf for suggestions regarding the constrained inverse kinematics. Funding: KJS: Werner Siemens Fellowship of the Swiss Study Foundation; PM: Smart Start I, Bernstein Center for Computational Neuroscience; MWM: the Rowland Fellowship from the Rowland Institute

at Harvard, and SNSF grant (310030_201057). AM: SNSF grant (310030_212516); AM and MWM funding from EPFL.

## Additional information

### Competing interests
Mackenzie Weygandt Mathis: Reviewing editor, eLife. The other authors declare that no competing interests exist.

### Funding

| Funder | Grant reference number | Author |
|---|---|---|
| Swiss National Science Foundation | 310030_201057 | Mackenzie Weygandt Mathis |
| Swiss National Science Foundation | 310030_212516 | Alexander Mathis |
| Rowland Institute at Harvard | | Mackenzie Weygandt Mathis |
| EPFL | | Alexander Mathis Mackenzie Weygandt Mathis |

The funders had no role in study design, data collection and interpretation, or the decision to submit the work for publication.

### Author contributions
Kai J Sandbrink, Software, Formal analysis, Validation, Investigation, Visualization, Methodology, Writing – original draft, Writing – review and editing; Pranav Mamidanna, Data curation, Software, Formal analysis, Investigation, Visualization, Methodology, Writing – original draft, Writing – review and editing; Claudio Michaelis, Methodology; Matthias Bethge, Resources, Supervision, Writing – review and editing; Mackenzie Weygandt Mathis, Conceptualization, Resources, Supervision, Funding acquisition, Validation, Visualization, Methodology, Writing – original draft, Project administration, Writing – review and editing; Alexander Mathis, Conceptualization, Resources, Formal analysis, Supervision, Funding acquisition, Visualization, Methodology, Writing – original draft, Project administration, Writing – review and editing

### Author ORCIDs

Pranav Mamidanna http://orcid.org/0000-0002-2095-3314
Matthias Bethge http://orcid.org/0000-0002-6417-7812
Mackenzie Weygandt Mathis http://orcid.org/0000-0001-7368-4456
Alexander Mathis http://orcid.org/0000-0002-3777-2202

### Decision letter and Author response
Decision letter https://doi.org/10.7554/eLife.81499.sa1
Author response https://doi.org/10.7554/eLife.81499.sa2

## Additional files

### Supplementary files
• MDAR checklist

### Data availability
The computational dataset and code to create it is available at https://github.com/amathislab/Deep-Draw, (copy archived at *Sandbrink et al., 2023*).

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
