## [Editor Report]

This article proposes a combination of biomechanical modeling and in-silico experiments, on a newly-curated passive-movement dataset, to elucidate the nature of computations in the proprioceptive pathway. The authors find that, in addition to its canonical role in representing the body state, the proprioceptive pathway may have evolved to recognize actions. Overall, the authors' findings lead to new hypotheses about proprioception that future in-vivo experiments could test.

---

## [Decision Letter]

**Decision letter after peer review:**

Thank you for submitting your article "Contrasting action and posture coding with hierarchical deep neural network models of proprioception" for consideration by *eLife*. Your article has been reviewed by 2 peer reviewers, and the evaluation has been overseen by a Reviewing Editor and Timothy Behrens as the Senior Editor. The following individual involved in the review of your submission has agreed to reveal their identity: Nikolaus Kriegeskorte (Reviewer #2).

The reviewers found the topic of the paper and its contributions interesting.

At the same time, overall, the reviewers

1. Expressed reservations at the lack of quantitative evaluation of neural data, and more importantly

2. Suggest that to lend support to the current claim that the function of the proprioceptive system is action recognition, the authors would need to train with losses intermediate between posture and action that, in addition to action can decode velocity as well.

The Reviewing Editor agrees with the reviewers that 2. would strengthen the manuscript. In its absence, the Reviewing Editor would find it difficult to fully support the scientific findings of the manuscript. The Reviewing Editor encourages the authors to consider additional experiments based on the detailed comments from the reviewers.

*Reviewer #1 (Recommendations for the authors):*

As I have explained in the public review, I don't agree with the main conclusion of the paper. The end effector position estimation task is too simple and is not the right model to contrast the action recognition model with. Ideally, I would argue that one should simulate a full closed-loop control model to be able to make a claim like this for proprioception. At the very least, they could have a state estimation network that estimates effector velocities in addition to position, which would likely learn speed, velocity, and direction-selective hidden layer representations. As far as I can see, the lack of such cells is the criterion that they use to rule out the TDT task.

I don't know if it exists, but if there is relevant data available, they could make more detailed comparisons to data. For example, do the trends they see in the change of selectivity across network layers match that in the brain?

*Reviewer #2 (Recommendations for the authors):*

1) The title refers to "posture", but this term is not at all mentioned in the abstract, which instead contrasts action and trajectory decoding. It would be good to choose consistent terminology or at least to relate the term posture to the trajectory decoding objective in the abstract.

2) Some more motivation is needed, even in the abstract, for the idea that the function of the proprioceptive pathway is to recognize what the brain already knows: the action it is trying to perform. This is fascinating and plausible to me, but it is not at all obvious. The introduction and discussion should contain the authors' best guesses (even if speculative) to better motivate the ART objective and interpret the central result.

3) Please address the two weaknesses I mention in the public review (no quantitative evaluation, unclear what about the ART drives the main result) in the discussion with a view to guiding future follow-up studies (including ones that build on the data set and code from this study).

4) It might be clearer to refer to the control models as untrained models (as different other kinds of control model could have been used).

5) Figure 4A: "CKA score" is not a conceptually informative label for the color bar. "similarity between untrained and trained model representations [CKA]" would be better.

6) Figure 4B: The t-SNE plots seem to show little more than that the ART, but not the TDT model shown achieves the ART model's objective. If models trained with different objectives are to be compared, then the visualization should not be partial to one of the objectives. Coloring dots by trajectory similarity might provide an alternative scheme that could be added. More generally, t-SNE requires setting hyperparameters and its objective function is hard to state concisely (beyond maintaining neighbor relationships), making interpretation more difficult than for say metric-scale MDS. t-SNE may also show classes as clustered that are not linearly separable, failing to visualize the gradual disentangling across layers, as in Figure 5 here: https://arxiv.org/abs/2107.00731. I have no definite suggestion but did not find this panel particularly informative in the present form.

7) Figure 4C: With one tick mark per letter, some readers will miss the many conditions per letter that are shown in the RDMs (despite this being apparent in the proprioceptive RDM). It might be better to show the whole matrix much larger in a separate supplemental figure and to focus on a, b, c, d, and e in Figure 4C and to add tickmarks for, say 5 instances for each of these letters. The number of conditions shown should also be stated in the figure legend and ideally should match the number of tickmarks on the matrix.

8) Figure 5E: The unconventional lines connecting percentiles across the vertical distribution plots add more clutter than clarity. It might be better to plot the distributions and point summaries separately or choose one. An exacerbating factor might be that the coding of the different tuning properties in the shading is not clearly discernible, and I end up having to count which of the five tuning variables I am looking at. Using a separate plot for each tuning type might help.

9) The paper contains some ungrammatical English and typos and would benefit from careful proofreading and editing.

10) CKA stands for "centered kernel alignment" (not "centered kernel analysis").

---

## [Author Response]

The reviewers found the topic of the paper and its contributions interesting.At the same time, overall, the reviewers1. Expressed reservations at the lack of quantitative evaluation of neural data, and more importantly2. Suggest that to lend support to the current claim that the function of the proprioceptive system is action recognition, the authors would need to train with losses intermediate between posture and action that, in addition to action can decode velocity as well.The Reviewing Editor agrees with the reviewers that 2. would strengthen the manuscript. In its absence, the Reviewing Editor would find it difficult to fully support the scientific findings of the manuscript. The Reviewing Editor encourages the authors to consider additional experiments based on the detailed comments from the reviewers.

We thank all reviewers for their insightful comments. Firstly, we agree overall that we should tone-down the perception that we claimed proprioception is doing action recognition at the cost of trajectory estimation. We don’t want to claim that – we want to simply put forth that normative models can yield hypotheses, and proprioception might do action recognition. Thus, firstly we tuned our writing. Also, as suggested by the reviewers, we conducted additional experiments to include an intermediate model that predicts location and velocity, as well. We find that this generalized TDT model (that also predicts velocity), still has much fewer directional selective units than the ART model (see Suppl. Figure 6-3). Thanks for this suggestion, we believe it did improve the manuscript.

Furthermore, we do agree that a comparison to neural data is of great interest. However, here we presented a model based on human biomechanical models. We thus feel that predicting neural data is beyond the scope of this paper (as we are not aware of single unit recordings of the proprioceptive pathway in humans). We do note in the revision that since the time of our preprint, one study did test our hypothesis and show cuneate nucleus has more intermediate-layer neurons[68], and we note this in the revision. And, over the past two years we have started follow up work with a Macaque arm model and are currently comparing data in the brain stem and sensory cortex, but this is much beyond the scope of this manuscript.

Reviewer #1 (Recommendations for the authors):As I have explained in the public review, I don't agree with the main conclusion of the paper. The end effector position estimation task is too simple and is not the right model to contrast the action recognition model with. Ideally, I would argue that one should simulate a full closed-loop control model to be able to make a claim like this for proprioception. At the very least, they could have a state estimation network that estimates effector velocities in addition to position, which would likely learn speed, velocity, and direction-selective hidden layer representations. As far as I can see, the lack of such cells is the criterion that they use to rule out the TDT task.

We thank reviewer #1 for their insightful comments. In this paper, we are intentionally focusing on proprioception as its own system (and not entangled with motor control). Much like vision is studied without motor (which notably changes visual properties) we aim to study this “sensory system" in isolation. We fully agree with you that studying proprioception in a closed-loop fashion (with a motor system) will be a key next step, but is beyond the scope of this manuscript, and our aim is not to overthrow the canonical role of proprioception, rather put forth a way to isolate and test hypotheses with a normative approach. We have toned-down the language and rather aim to highlight here is an example (i.e., propose proprioception could involve action recognition, much like vision can also do face recognition). We hope this revision clarifies our intent better.

Furthermore, we believe that there is value in studying proprioception in isolation to later contrast the results for closed-loop system. To emphasize this limitation, we have updated the introduction (and discussion), where we state: “We emphasize, that as in previous task-driven work for other sensory systems (Yamins and DiCarlo 2016, Kell et al. 2018), we do not model the closed-loop nature. Of course, in reality, the motor system is not constant but the tuning properties of proprioception are optimized jointly with the motor system and other changing influences. Studying proprioception with a basic open-loop model is important to set the stage for more complex models such as joint models of proprioception and motor control."

Our goal in this paper is not to assert that either trajectory decoding or action recognition is the "true" or even the "canonical" model of the proprioceptive system. Rather, we view trajectory decoding as representative of the set of tasks whose goal would be to recover information that can be used by the rest of the brain to accomplish higher-order functions. We also note that we added another trajectory decoding task as a control (see below). We select action recognition as a paradigmatic case of one of these higher-order cognitive tasks (although we by no means view it as the true, or only, function of the proprioceptive system). Our aim by doing this is to show that training a network on a higher-order task is sufficient for recovering the kinds of neural representations that have been observed in biological measurements of the proprioceptive system. We adjusted the writing in the paper to make this conceptual distinction clearer.

We agree that the comparison between the two tasks could be enriched by including velocity prediction as task for the artificial proprioceptive system. As suggested by the reviewer we also added a new task predicting both the position and the velocity of the end-effector. Interestingly, we also do not find many direction selective units in this case (in comparison to the purely positional TDT,see Suppl. Figure 6-3) and indeed also this task unlearns direction selectivity in comparison to the untrained initializations. The ART-trained models have significantly more direction selective units.

I don't know if it exists, but if there is relevant data available, they could make more detailed comparisons to data. For example, do the trends they see in the change of selectivity across network layers match that in the brain?

As we noted in the response to both reviewers, we are not aware of human-recordings in the proprioceptive system that are available. We believe that comparing directly to neural data is beyond the scope of the current manuscript.

In lieu of that, we compare the emergent representations in the networks with neural data qualitatively where this is appropriate, and indeed some findings have already been validated experimentally now that our pre-print has been available for 3 years (as in the case of the distribution of preferred directions).

Reviewer #2 (Recommendations for the authors):1) The title refers to "posture", but this term is not at all mentioned in the abstract, which instead contrasts action and trajectory decoding. It would be good to choose consistent terminology or at least to relate the term posture to the trajectory decoding objective in the abstract.

Thanks for that observation, we have now utilized posture in the abstract and the introduction. There we state: “One key role of proprioception is to sense the state of the body—i.e., posture. This information subserves many other functions, from balance to motor learning. [..]"

2) Some more motivation is needed, even in the abstract, for the idea that the function of the proprioceptive pathway is to recognize what the brain already knows: the action it is trying to perform. This is fascinating and plausible to me, but it is not at all obvious. The introduction and discussion should contain the authors' best guesses (even if speculative) to better motivate the ART objective and interpret the central result.

Thanks for this feedback. As described in the response to Reviewer 1 above, we do not take our work in this paper to imply that action recognition is the "true" model of the proprioceptive system. Rather, we consider it as a paradigmatic case of one of the (many) higher-order cognitive tasks that the proprioceptive system can give access to. Our aim by doing this is to show that training a network on a higher-order task is sufficient for recovering the kinds of neural representations that have been observed in biological measurements of the proprioceptive system. We will adjust the writing in the paper to make this conceptual distinction clearer.

3) Please address the two weaknesses I mention in the public review (no quantitative evaluation, unclear what about the ART drives the main result) in the discussion with a view to guiding future follow-up studies (including ones that build on the data set and code from this study).4) It might be clearer to refer to the control models as untrained models (as different other kinds of control model could have been used).

Thanks for the suggestion, we call it untrained model now.

5) Figure 4A: "CKA score" is not a conceptually informative label for the color bar. "similarity between untrained and trained model representations [CKA]" would be better.

Thanks for the comment, we changed the label.

6) Figure 4B: The t-SNE plots seem to show little more than that the ART, but not the TDT model shown achieves the ART model's objective. If models trained with different objectives are to be compared, then the visualization should not be partial to one of the objectives. Coloring dots by trajectory similarity might provide an alternative scheme that could be added. More generally, t-SNE requires setting hyperparameters and its objective function is hard to state concisely (beyond maintaining neighbor relationships), making interpretation more difficult than for say metric-scale MDS. t-SNE may also show classes as clustered that are not linearly separable, failing to visualize the gradual disentangling across layers, as in Figure 5 here: https://arxiv.org/abs/2107.00731. I have no definite suggestion but did not find this panel particularly informative in the present form.

We agree that the criticism of t-SNE is well taken, and we only use it as a visualization. The gradual disentanglement is quantified with RDMs in the same figure (4D-F).

However, we respectfully disagree with the reviewer. Of course the TDT model does not disentangle the characters, but we have found this panel a useful illustration for some readers/listeners. That said, we do agree, it’s not very deep.

7) Figure 4C: With one tick mark per letter, some readers will miss the many conditions per letter that are shown in the RDMs (despite this being apparent in the proprioceptive RDM). It might be better to show the whole matrix much larger in a separate supplemental figure and to focus on a, b, c, d, and e in Figure 4C and to add tickmarks for, say 5 instances for each of these letters. The number of conditions shown should also be stated in the figure legend and ideally should match the number of tickmarks on the matrix.

That’s a great idea, we made this change.

8) Figure 5E: The unconventional lines connecting percentiles across the vertical distribution plots add more clutter than clarity. It might be better to plot the distributions and point summaries separately or choose one. An exacerbating factor might be that the coding of the different tuning properties in the shading is not clearly discernible, and I end up having to count which of the five tuning variables I am looking at. Using a separate plot for each tuning type might help.

We thank the reviewer for this comment, we have dropped the percentile lines and adjusted the colors and labels for clarity.

9) The paper contains some ungrammatical English and typos and would benefit from careful proofreading and editing.

We have carefully read the paper again and hopefully fixed all errors.

10) CKA stands for "centered kernel alignment" (not "centered kernel analysis").

Thank you, we have changed the text accordingly.